

# Climatic and Tectonic Forcing Lead to Contrasting Headwater Slope Evolutions

Yinbing Zhu[1], Patrice Rey[1], Tristan Salles[1]

[1]School of Geosciences, University of Sydney, Sydney, NSW, 2006, Australia

*Correspondence to*: Patrice F. Rey (patrice.rey@sydney.edu.au)

**Abstract.** Landscapes evolve through the coupled effects of tectonics and surface processes. Previous studies have shown that uplift rate changes generate upstream-migrating erosion waves, altering downstream slopes while upstream ones remain constant until the wave arrives. However, the distinctive differences between landscape responses to uplift versus climatic changes, particularly rainfall rate changes, remain incompletely described. This study uses a numerical model to investigate

landscape responses to changes in both rainfall and uplift rates. Results show that, unlike the simple upstream-migrating erosion waves from uplift rate changes, rainfall rate changes generate more complex responses. Specifically, rainfall rate changes cause transient slope change reversals at the headwaters due to differential erosion between the divide and its adjacent areas, a pattern not observed in uplift-induced evolution. These reversals are more pronounced when hillslope diffusion plays a dominant role (i.e., high diffusion coefficient). While both tectonic and climatic forcing drive landscape

change, they produce recognizably different signatures in river profiles. If these distinctive signatures can be identified from river profiles or inferred from erosion rate measurements, they can help disentangle climatic and tectonic influences on landscape evolution.

## 1 Introduction

Whilst tectonic and geodynamic forces generate longer wavelength topography, Earth's surface processes powered by

climate dissect the Earth's surface, creating high-frequency topographic features that contribute to the reconfiguration of drainage patterns and the re-routing of sediments from source to sink (e.g., Allen, 2008; Wobus et al., 2006a; Whipple et al., 2013; Martinsen et al., 2022; Seybold et al., 2021). Whether or not climatic and tectonic disturbances impact landscape evolution differently has been debated for decades (e.g., Kirby and Whipple, 2012; Whipple, 2009; Bonnet and Crave, 2003; Whittaker, 2012). Previous research has focused on various landscape features, such as river channels, drainage divide, and

alluvial fans, to understand whether or not they respond differently to tectonic and climatic disturbances (Leonard and Whipple, 2021; Mao et al., 2021; Shi et al., 2021; Willett et al., 2014). Rivers, in particular, have been found to respond strongly to climatic and tectonic disturbances, making them a valuable feature for studying how landscapes evolve (Molin et al., 2023; Quye-Sawyer et al., 2021; D'arcy and Whittaker, 2014). Here, we investigate via numerical experiments how river channels respond to climatic and uplift disturbances, paying particular attention to the role of hillslope diffusion, which is



often overlooked in favour of river incision processes. We show that river channels respond slightly differently to tectonic and climatic changes when hillslope diffusion is considered. After changes in uplift rate, the channel slope at the headwaters records a monotonic increase (uplift rate increase) or decrease (uplift rate decrease). In contrast, after changes in rainfall rate, the channel slope records a non-monotonic adjustment, which becomes more pronounced as the surface diffusion coefficient increases. We suggest that changes in rainfall rate cause a transient spatial variation in erosion rate around the divide area

due to the interaction between hillslope diffusion and river incision. This difference has the potential to distinguish between tectonic and climatic influences on landscape evolution.

**1.1 River incision vs hillslope diffusion**

Several numerical models have been proposed to quantify river incision processes (e.g., Dietrich et al., 2003; Howard and Kerby, 1983; Perron et al., 2008). The most commonly used is the detachment-limited stream power model, which assumes

that sediments are instantly flushed from the channel and that the bedrock erosion rate $E$ depends on the channel slope $S$, drainage area $A$, and precipitation $P$:

$$E = k_d (PA)^m S^n \tag{1}$$

where $m$ and $n$ are positive constant exponents, and $k_d$ is a coefficient describing the erodibility of the channel bed and reflects the combined impacts on the erosion of climate, lithology, bedload, and other potential parameters (Kirby and

Whipple, 2012; Smith et al., 2022; Whipple and Tucker, 1999). Following the principle of conservation of mass, the rate of surface elevation change ($\partial z / \partial t$) is determined by the difference between the uplift rate $U$ and erosion rate:

$$\frac{\partial z}{\partial t} = U - E \tag{2}$$

As rivers incise, the sloping ground at their flanks increases, driving hillslope diffusion, which describes the downward transport of creeping soil (Fernandes and Dietrich, 1997; Dietrich et al., 2003). Models indicate that the convexity of the

hillslope profile is influenced by hillslope processes and the rate of incision at the hillslope base (e.g., Armstrong, 1987; Ahnert, 1987). Hence, river incision and hillslope diffusion are coupled and evolve simultaneously. A simple model describing the process of hillslope diffusion assumes that the flux of soil along hillslopes is linearly related to the hillslope gradient (e.g., Culling, 1963, 1960; Salles and Duclaux, 2014; Tucker and Hancock, 2010):

$$\frac{\partial z}{\partial t} = k_{hl} \nabla^2 z \tag{3}$$

where $k_{hl}$ is the hillslope diffusion coefficient, which integrates climate, lithology, soil conditions, and biotic influences (Dietrich and Perron, 2006; Hurst et al., 2013; Robl et al., 2017). Hillslope diffusion gradually transports soil and sediment downslope due to gravity and reshapes substantially the landscape over time (e.g., Litwin et al., 2024; Perron et al., 2008; Roering, 2008). It has been shown that hillslope diffusion strongly influences drainage density and valley spacing (Perron et al., 2008; Sweeney et al., 2015; Tucker and Bras, 1998). Additionally, the sediment and soil transported from hillslopes

impact river incision by either acting as tools for erosion or forming a protective cover that shields the underlying bedrock from further erosion (Sklar and Dietrich, 2001).



While much research has focused on river channel evolution (e.g., Kirby and Whipple, 2012; Wobus et al., 2010), few have explored whether and how river channels respond differently to tectonic and climatic changes when hillslope diffusion is included. Before addressing this issue, the following paragraph clarifies the notions of steady-state and transient landscapes.

**1.2 Steady state vs transient landscapes**

Computer-generated landscapes evolving under controlled tectonic and climatic conditions provide a robust framework for better understanding the formation and evolution of natural landscapes (e.g., Chen et al., 2014; Pan et al., 2021; Salles and Hardiman, 2016; Schwanghart and Scherler, 2014). These models show that a landscape reaches a steady state when the uplift rate equals the erosion rate. When the uplift rate changes, landscapes are in a transient state of disequilibrium and
evolve to reach a new steady state (e.g., Leonard and Whipple, 2021; Miller et al., 2012; O'hara et al., 2019). Steady-state and transient landscapes show a sharp contrast in the variation of rivers' profiles. When a river channel has reached a steady state, its longitudinal elevation profile is usually smooth and concave-up (Fig. 1a). In contrast, under uniform lithology, knickpoints form in transient river channels (Wobus et al., 2006b; Lague, 2014; Neely et al., 2017; Whipple et al., 2013). A knickpoint is a location where there is an abrupt change in the channel slope (Fig. 1b). A positive knickpoint forms where
the slope suddenly increases downstream, while a negative knickpoint forms where the slope decreases abruptly. A mobile positive knickpoint indicates an increase in uplift rate and/or a decrease in erosion efficiency (induced by a decrease in rainfall rate, for example), while a mobile negative knickpoint indicates the opposite conditions (Baldwin et al., 2003). Both types of knickpoints typically form at the river mouth and migrate upstream toward the headwaters.

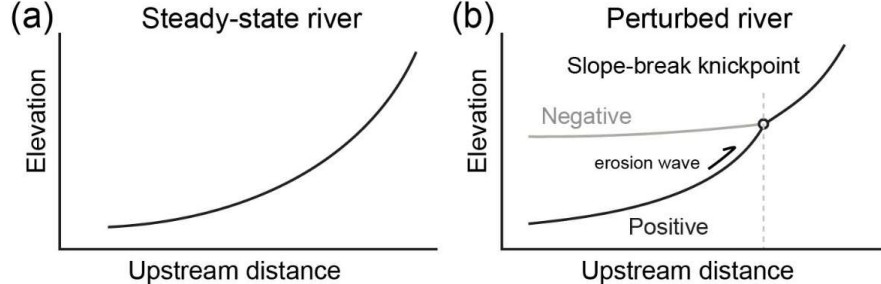


**Figure 1. Channel profiles with different morphology. (a) a steady-state river profile. (b) Transient river profiles with a negative or positive slope-break knickpoint.**

A migrating knickpoint separates the channel into two segments, upstream and downstream segments. It has been proposed
that regardless of whether the transient change is driven by tectonics or climate, the elevation of the upstream segment changes while its slope remains constant. After the downstream segment reaches a steady state, its channel elevation and slope have changed (e.g., Whipple, 2001; Whipple and Tucker, 1999).



**2 Methodology and model setup**

To investigate landscape evolution under climatic or tectonic changes, as well as varying erodibility and hillslope diffusion,
we use the long-term surface evolution model Badlands (Basin and Landscape Dynamics) (Salles, 2016; Salles and
Hardiman, 2016). Badlands is designed to simulate i/ landscape development via the mobilisation of sediments through
hillslope diffusion and stream-power incision, ii/ sediment transport from source to sink and into marine environments, iii/
sediment accumulation in sedimentary basins, and iv/ isostatic re-adjustment of Earth's lithosphere due to surface loading
and unloading. Although nonlinear diffusion models may better describe sediment transport on hillslopes (e.g., Jiménez-
Hornero et al., 2005; Martin, 2000; Roering et al., 1999), for simplicity, our model assumes that hillslope sediment transport
rates are linearly proportional to the slope gradient. Here, we explore landscape responses to changes in rainfall or uplift, and
we disregard isostatic re-adjustment. In particular, we focus on contrasts in drainage network patterns, contrasts in average
elevation, contrasts in surface roughness, and contrasts in river profiles.

Our initial landscape models are mapped over a 40 km × 80 km grid with a uniform initial elevation of 10 m and a spatial
resolution of 400 m × 400 m. We design four initial models with varying hillslope diffusion and erodibility coefficients
(Table 1). The diffusion coefficient is set to 0 in model M1, meaning the landscape evolution is purely driven by riverine
processes with an erodibility coefficient of $2.3 \times 10^{-6}$ $yr^{-1}$. We set the diffusion coefficient to 1 $m^2$/yr in model M2 and 2
$m^2$/yr in model M3. Finally, in our last model M4, the erodibility is doubled to $4.6 \times 10^{-6}$ $yr^{-1}$.

**Table 1. Diffusion coefficient and erodibility of four models**

| Model | Diffusion coefficient $k_{hl}$ (m²/yr) | Erodibility $k_d$ (1/yr) |
|-------|----------------------------------------|--------------------------|
| M1    | 0                                      | $2.3 \times 10^{-6}$     |
| M2    | 1                                      | $2.3 \times 10^{-6}$     |
| M3    | 2                                      | $2.3 \times 10^{-6}$     |
| M4    | 2                                      | $4.6 \times 10^{-6}$     |


Our four models are submitted to a combination of uniform uplift at a rate of 300 m/Myr and background rainfall at a rate of
2 m/yr until they reach a steady state equilibrium, where mean elevation and river profiles no longer change (Montgomery,
2001; Willett & Brandon, 2002). This first stage lasts for 25 Myr (Fig. 2), after which all models reach a steady state.

In the second stage, which also lasts 25 Myr, each model is subjected to a perturbation while the other forcing remains
constant. We either:

- Increase rainfall to 6 m/yr or decrease it to 0.67 m/yr, while keeping uplift fixed at 300 m/Myr, or
- Increase uplift to 900 m/Myr or decrease it to 100 m/Myr, while keeping rainfall fixed at 2 m/yr.



This design yields 16 individual experiments (Fig. 2), allowing us to assess landscape responses to changes in rainfall and uplift rates separately.


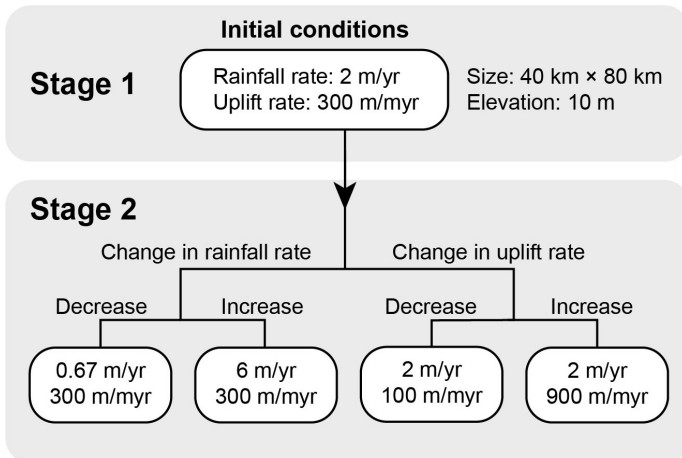

**Figure 2. Each of our four initial models (M1 to M4) experiences four different two-stage landscape evolutions controlled by changes in rainfall or uplift. Stage 1: An initial flat landscape is uplifted under an uplift rate of 300 m/Myr and a rainfall rate of 2 m/yr until a steady-state landscape is reached. Stage 2: Changes in rainfall or uplift rate.**

**3 Results**

**3.1 Comparison of final, steady-state landscapes**

*Impact on patterns of drainage networks*: Despite having different erosion and diffusion coefficients and going through different climatic and tectonic histories, our four initial models display broadly similar patterns of drainage networks. In all 16 cases, the two largest drainage basins form at the eastern and western parts of the landscape, separated by a central divide
(Fig. 3). Interestingly, the drainage patterns in models M2 and M4 are highly similar, reflecting that both models have the same ratio of hillslope diffusion to erodibility.

*Impact on average elevation and surface roughness*: Our results show that the mean landscape elevation and surface roughness increase following a decrease in rainfall rate or an increase in uplift rate and decrease following an increase in rainfall rate or a decrease in uplift rate. Regardless of rainfall or uplift changes, the absence of hillslope diffusion in M1 ($k_{hl}$
$= 0$) leads to the largest surface roughness (Fig. 3a). When hillslope diffusion is included, the landscapes in models M2, M3, and M4 are smoother than those in model M1 (Fig. 3b-d). Models M2 and M4 show that, regardless of rainfall or uplift changes, doubling both the diffusion and erosion coefficients reduces both the mean elevation and the mean surface roughness by a factor of ~2. Interestingly, models M2 and M3 show that doubling only the diffusion coefficient reduces the



surface roughness by ~15% and, surprisingly, increases the mean elevation by ~20%. Models M3 and M4 show that
doubling the erosion coefficient alone reduces the mean elevation by a factor of more than 2.

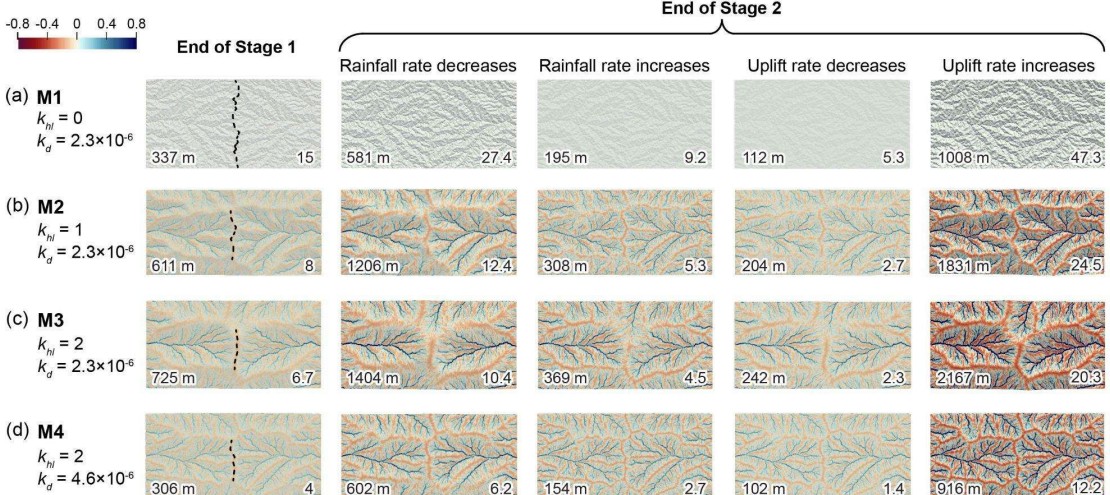

**Figure 3. Hillshade maps showing erosion and deposition rates resulting from hillslope diffusion at the end of Stage 1 and the end of Stage 2 for models M1 (a), M2 (b), M3 (c), and M4 (d). Each model differs in hillslope diffusion coefficients ($k_{hl}$) and erodibility values ($k_d$). Blue areas indicate deposition, while red areas represent erosion. Color bar values indicate depositional (positive) and erosional (negative) rates (mm/yr). Numbers at the bottom left of each map display the mean elevation, while numbers at the bottom right display the mean surface roughness, calculated using the 'roughness' algorithm of GDAL in QGIS (Wilson et al., 2007). Dashed lines on maps at the end of Stage 1 denote the divides. The divides in Stage 2 are similar to those in Stage 1 and are not marked in this stage.**

### 3.2 Impact on rivers' channel response

To explore channel responses to changes in rainfall or uplift rates under various ratios of hillslope diffusion to erodibility, we analyze the trunk stream of the western basin, including the evolution of erosion and deposition, as well as the evolution of the longitudinal channel profile. Although we present results only from the western basin, we have verified that both drainage basins exhibit similar evolutions. Given the overall consistency in river channel behavior across all models, we present the results from model M1 (no hillslope diffusion), highlighting the key differences observed in models M2 to M4 (all with non-zero hillslope diffusion).

After a decrease in rainfall rate or an increase in uplift rate, the trunk stream is uplifted with the increase in landscape elevation and the channel slope. A positive knickpoint and an erosion wave develop, migrating upstream from the river mouth (Fig. 4a and d). The downstream channel first reaches a steady state, with no further changes in elevation and slope. The entire channel returns to a new steady state once the erosion wave reaches the headwaters and the knickpoint disappears.




Conversely, an increase in rainfall rate or a decrease in uplift rate lowers the channel elevation and its slope, creating a negative knickpoint at the river mouth and an erosion wave that migrates upstream (Fig. 4b and c). Once the erosion wave reaches the headwaters, the channel eventually reaches a steady state.


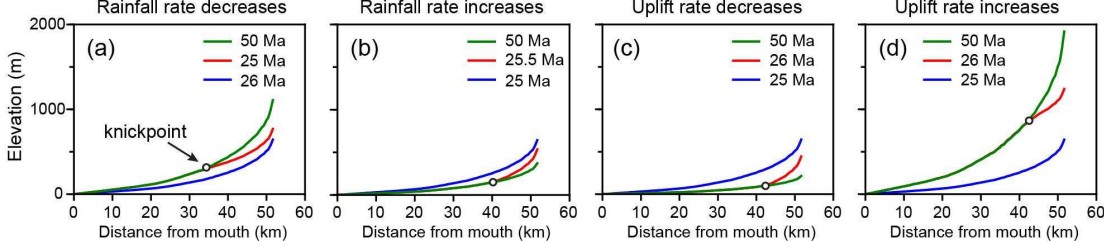

**Figure 4. Longitudinal profiles of the trunk stream after changes in rainfall or uplift rates in model M1. The changes occur at 25 Ma, affecting the steady state trunk stream in blue.**

When the hillslope diffusion is absent (model M1), the slope of the trunk stream headwaters remains nearly constant for 1-2 Myrs following a change in either uplift or rainfall rate (Fig. 5 a1-3 and Fig. 6 a1-3). The slope then increases monotonously after the arrival of the upstream migrating erosion wave (Whipple and Tucker, 1999).



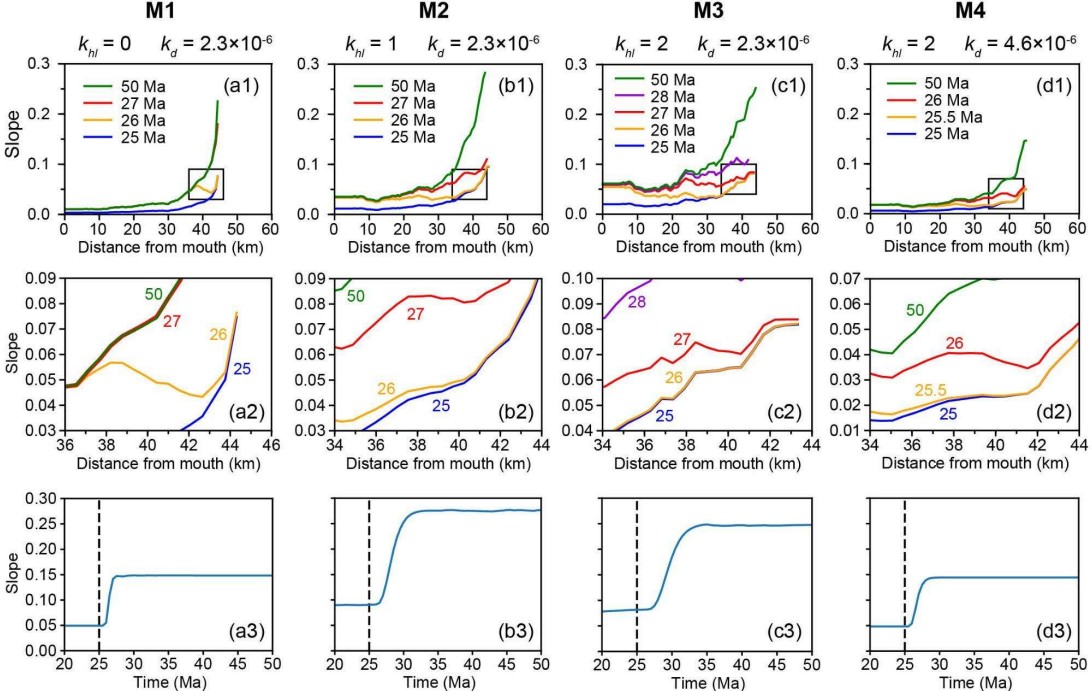

**Figure 5. Evolution of trunk stream slope following an increase in uplift rate. (a1-d1) Longitudinal slope profiles of the trunk stream at selected time steps (colored lines), with each subplot corresponding to a model (M1-M4). Black rectangles indicate the headwater regions. (a2-d2) Enlarged views of the headwater areas, corresponding to the boxed regions in (a1-d1). (a3-d3) Temporal evolution of the mean channel slope in the upper ~800 m of the trunk stream, capturing the dynamic slope response across model runs. Dashed vertical lines mark the timing of the uplift rate increase (25 Ma).**





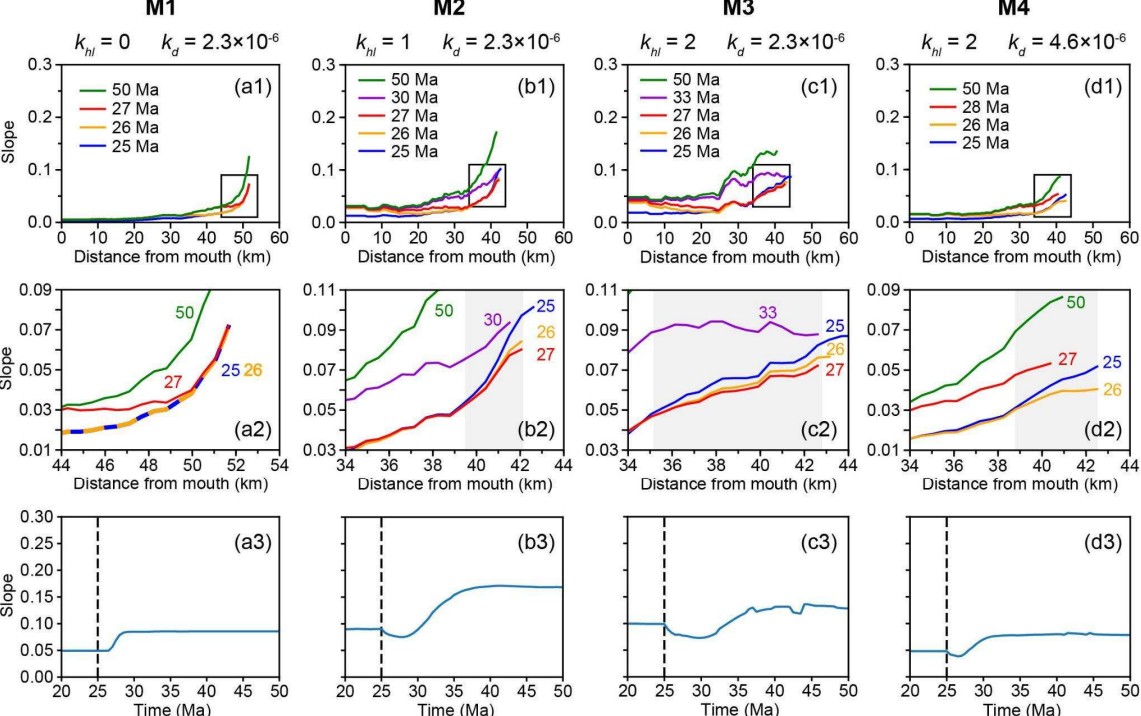

**Figure 6. Evolution of trunk stream slope following a decrease in rainfall rate. (a1-d1) Longitudinal slope profiles of the trunk stream at selected time steps (colored lines), with each subplot corresponding to a model (M1-M4). Black rectangles indicate the headwater regions. (a2-d2) Enlarged views of the headwater areas, corresponding to the boxed regions in (a1-d1). Grey bands**
**indicate the regions where the transient slope change reversal occurs. (a3-d3) Temporal evolution of the mean channel slope in the upper ~800 m of the trunk stream, capturing the dynamic slope response across model runs. Dashed vertical lines mark the timing of the rainfall rate decrease (25 Ma).**

In contrast, when hillslope diffusion is present (models M2, M3, and M4), we observe major differences in the evolution of

river headwater slopes following changes in uplift and rainfall rates. An increase in uplift rate leads to a monotonic slope increase in the headwaters (Fig. 5 b1-3, c1-3, and d1-3). In contrast, a decrease in rainfall rate first leads to a transient slope decrease, followed by a subsequent increase, a phenomenon we refer to as "*transient slope change reversal*" (Fig. 6 b1-3, c1-3, and d1-3). The opposite pattern occurs when the rainfall rate increases: a temporary slope increase is followed by a decrease. Interestingly, we find that transient slope change reversals are associated with hillslope diffusion and river incision.

In model 3, which has the largest ratio of diffusion coefficient to erodibility, longer channels experience transient slope change reversals (Fig. 6 c2), and transient slope change reversals persist longer in time than in the other models (Fig. 6 c3).

In contrast, when hillslope diffusion is present (models M2, M3, and M4), we observe major differences in the evolution of river headwater slopes following changes in uplift and rainfall rates. An increase in uplift rate leads to a monotonic slope





increase in the headwaters (Fig. 5 b1-3, c1-3, and d1-3). In contrast, a decrease in rainfall rate first leads to a transient slope

decrease, followed by a subsequent increase, a phenomenon we refer to as "*transient slope change reversal*" (Fig. 6 b1-3, c1-3, and d1-3). The opposite pattern occurs when the rainfall rate increases: a temporary slope increase is followed by a decrease. Interestingly, we find that transient slope change reversals are associated with hillslope diffusion and river incision. In model M3, which has the largest ratio of diffusion coefficient to erodibility, longer channels experience transient slope change reversals (Fig. 6 c2), and transient slope change reversals persist longer in time than in the other models (Fig. 6 c3).

**4 Discussion**

To better understand the cause of the transient slope change reversal, we calculate the erosion rate for each grid cell 1 Myr after the disturbance and extract the erosion rate along the trunk stream for all models (Fig. 7). The transient slope change reversal is driven by differential erosion rates between the divide and adjacent areas.

In model M1, the erosion rates of the divide and its adjacent areas remain homogeneous following changes in rainfall and
uplift rates (Fig. 7 a3). Similarly, in models M2, M3, and M4, an increase or decrease in uplift rate results in consistent erosion rates between the divide and adjacent areas (red and orange profiles in Fig. 7 b3, c3, and d3). The surface uplift rate is defined as the difference between the uplift and erosion rates. Given the spatial uniformity of uplift rates, equal erosion rates at the divide and its adjacent areas result in identical surface uplift rates, preventing transient slope change reversals (black and red profiles in Fig. 8).

In contrast, following a decrease in rainfall rate in models M2, M3, and M4, the erosion rate of the divide exceeds that of adjacent downstream areas (green profiles in Fig. 7 b3, c3, and d3). This difference in erosion rate directly causes the surface uplift rate of the divide to be lower than that of adjacent downstream areas, resulting in a temporary decrease in the channel slope at the divide and, therefore, triggering a transient slope change reversal (green profile in Fig. 8). Conversely, following an increase in rainfall rate, the erosion rate of the divide is lower than in adjacent areas (blue profiles in Fig. 7 b3, c3, and
d3), causing a temporary slope increase at the divide and again triggering a transient slope change reversal (blue profile in Fig.8). These findings suggest that rainfall changes distinctly influence divide erosion patterns, with spatial contrasts in erosion rate playing a key role in driving transient slope responses.





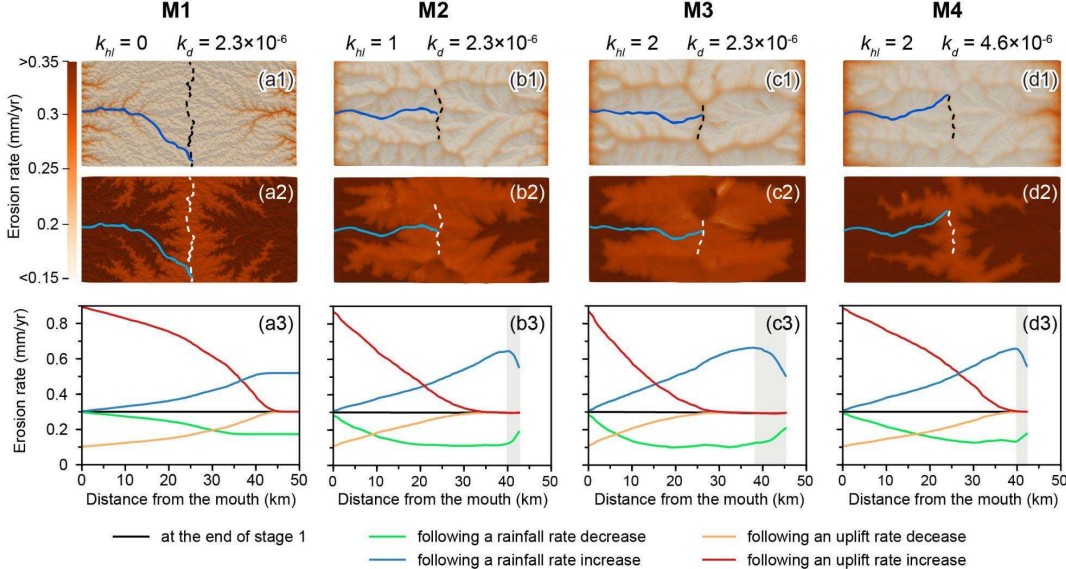

**Figure 7.** Erosion rates (mm/yr) per grid cell, calculated over 1 Myr following (a1-d1) a decrease in rainfall rate and (a2-d2) an increase in uplift rate. Blue lines in (a1-d1) and (a2-d2) represent trunk streams, and dashed lines mark divides. (a3-d3) Longitudinal erosion profiles along trunk streams, with grey bands indicating the regions where the transient slope change reversal occurs.

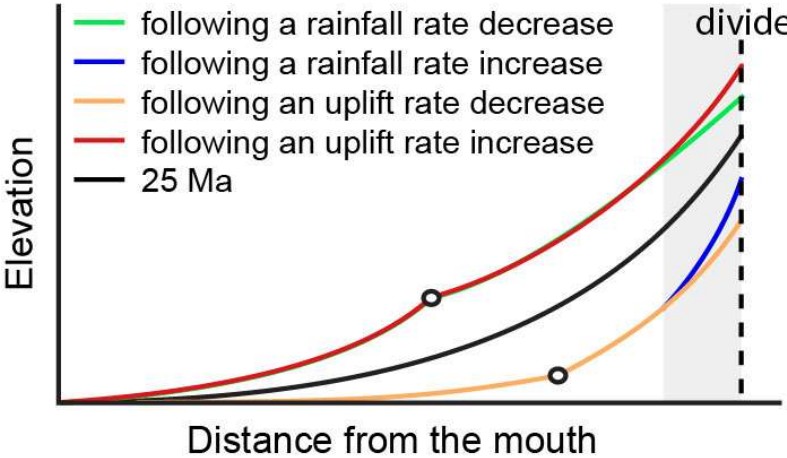

**Figure 8.** Schematic diagram of the longitudinal profile of the channel in a steady state (black line) or a transient state after changes in rainfall or uplift rate. The grey band indicates the region where the transient slope change reversal occurs.





Therefore, the interaction between hillslope diffusion and river incision is critical in understanding transient slope change

reversal. When the rainfall rate changes, the erosion rate throughout the channel is altered, leading to channel adjustment. However, hillslope response lags behind channel response (Clubb et al., 2019), as hillslope diffusion continues transporting materials downstream at the same rate as before the rainfall rate change. While river valleys are heavily influenced by incision, divides are dominated by hillslope diffusion (Dietrich et al., 2003).

Notably, increasing the hillslope diffusion coefficient results in smoother and wider divides and extends the channel length

affected by transient slope change reversals (Fig. 7 b3 and c3). In contrast, increasing the erodibility of bedrock enhances river incision. It weakens the diffusion influence on channel evolution, resulting in a narrower divide and a shorter channel reach where the transient slope change reversal occurs (Fig. 7 c3 and d3).

In summary, the transient slope change reversal results from the competition between incision and diffusion following a change in rainfall. This reversal disappears as the erosion wave gradually approaches the divide area, and the landscape

returns to a steady state where the erosion rate is spatially uniform.

Transient slope change reversals could be identified using slope-area analysis or χ analysis. Both methods rely on the stream power model, which describes the relationship between channel slope and drainage area as a power function (Flint, 1974). For a river channel in a steady state, plotting log slope against log area yields a straight line. However, in cases of transient slope change reversals, this relationship may deviate from linearity. While slope-area analysis can be sensitive to data noise

(e.g., DEM inaccuracies), χ analysis reduces this influence through an integral approach (Royden and Taylor Perron, 2013; Perron and Royden, 2013). For steady-state rivers, χ should also correlate linearly with elevation, whereas nonlinear χ-elevation relationships may indicate transient slope change reversals.

Transient slope change reversals could also be identified by investigating the erosion rate. One approach to quantify erosion rates is using cosmogenic nuclides, particularly radionuclides like $^{10}Be$ and $^{26}Al$ (e.g., Balco et al., 2008; Gosse and Phillips,

2001; Lal, 1991; Muzikar, 2009). These nuclides are produced in surface minerals by cosmic ray interactions, with production rates decreasing exponentially with depth due to cosmic ray attenuation (Dunai, 2010; Lal, 1991). Cosmogenic nuclide concentrations increase as a surface remains exposed to cosmic rays (Ivy-Ochs and Kober, 2008). In contrast, in rapidly eroding areas, nuclide concentrations remain low due to the continuous removal of surface materials.

By mapping nuclide concentrations, spatial patterns in erosion rates could be linked to rainfall or uplift changes. For

instance, if the erosion rate is relatively uniform around the divide area, it may suggest a transient response driven by tectonic events. Conversely, if nuclide data indicate increasing erosion at the divide while upstream erosion rates decline, a recent decrease in rainfall rate may be involved in the landscape evolution. Thus, cosmogenic nuclide measurements provide a valuable tool to distinguish between climatic and tectonic drivers of landscape change.



**5 Conclusion**

Changes in rainfall and uplift rates induce different responses in the channel slope at the headwaters, with hillslope diffusion playing a crucial role in adjusting these processes. When the rainfall rate changes, hillslope diffusion interacts with river incision to generate transient spatial variations in erosion around the divide area, leading to transient slope change reversals at the headwaters. In contrast, changes in uplift rates result in spatially uniform erosion across the divide area, preventing such reversals. Identifying these reversals from river profiles or erosion rate estimates at different locations could help

determine the driving force behind landscape adjustments. A high hillslope diffusion coefficient increases both the duration and spatial extent of these reversals along the river profile. In contrast, higher erodibility enhances river incision and diminishes the role of diffusion, reducing these reversal effects.

Our findings provide new insights into how climatic and tectonic forcing reshape landscapes over time. By investigating the interaction between diffusion and incision, we show that the transient variations in channel profiles, particularly near the

divide, provide potential markers for interpreting past landscape evolution and deciphering the complex interplay between tectonic uplift and climatic variability.

**Code and data availability.** Version 2.2.0 of Badlands used for the landscape and sedimentary evolution modeling is preserved at https://doi.org/10.5281/zenodo.1069573 (Salles & Howson, 2017), available via GNU General Public License

v3.0 and developed openly at https://github.com/badlands-model/badlands.

**Author contributions.** YZ designed and ran the simulations, analyzed the results, and wrote the manuscript. PR contributed to the result analysis and manuscript revision. TS developed the model code and contributed to the manuscript revision.

**Competing interests.** The authors declare that they have no conflict of interest.

**Acknowledgments.** The first author gratefully acknowledges the financial support from the China Scholarship Council (CSC) and the School of Geosciences at the University of Sydney.

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
