# Peer review of "Climatic and Tectonic Forcing Lead to Contrasting Headwater Slope Evolutions"

_EGUsphere, 2025_

## Referee Comment (RC1)

The manuscript is well-organized, clearly written, and presents a rigorous investigation into how landscapes respond to tectonic and climatic forcing, with a focus on slope evolution at river headwaters. Using the Badlands landscape evolution model, the authors demonstrate that changes in rainfall rate, unlike uplift rate changes, can induce transient slope change reversals near drainage divides, a process strongly modulated by hillslope diffusion. This insight offers a novel perspective on differentiating between climatic and tectonic controls in geomorphic systems. The experimental design is systematic, the interpretation is robust, and the study is presented in fluent and technically precise language. The work makes a valuable contribution to Earth surface process research and is suitable for publication with minor revisions. To further improve the clarity and scientific strength of the paper, the following suggestions are offered:

1. The authors argue that the transient slope change reversals are driven by differential erosion rates between the divide and adjacent areas. It is recommended to elaborate on why changes in rainfall rate lead to this contrast, while changes in uplift do not. What are the respective roles of hillslope diffusion and river incision in generating or amplifying these differences? A more detailed mechanistic discussion would strengthen the internal logic of the manuscript.

2. Since the authors mention that slope-area and chi analysis can help identify transient slope change reversals, it is recommended to include representative plots from the numerical models. These visualizations would help illustrate how such plots reveal the geomorphic response to climatic or tectonic perturbations.

3. The manuscript suggests that erosion rate patterns near drainage divides, measurable through cosmogenic nuclides, may help distinguish between climatic and tectonic forcing. Including a brief geological example would help bridge the model findings with field-based applications and strengthen the practical relevance of the study.

4. In the methodology section, it is recommended that the authors clearly list all model parameters used, including the values of m and n in the stream power incision equation. Additionally, the initial condition is described as a uniform elevation of 10 meters; however, the landscape at the end of the first stage appears asymmetric. If any initial perturbation or topographic noise was introduced to generate this asymmetry, it should be explicitly stated and justified in the methods section.

5. Technical Corrections
Figure 4(a): The labels for 25 Ma and 26 Ma are reversed in the figure legend.

---

## Referee Comment (RC2)

The manuscript is well organized, and the topic is attractive. In the Introduction section, the authors made a good summary of the work of their predecessors, and then put forward their own new understanding through the method of numerical simulation. The method is reliable. I read the manuscript with great interest as it is full of important knowledge in the field of drainage system evolution. However, after careful reading and consideration, I think there are two significant issues which need to be addressed before the possible publication of this manuscript.

Major concerns:
(1) Why the mean elevation increase when the hillslope diffusion increase? This is counterintuitive. In Line 134, the authors also consider it as an astonishing phenomenon. However, the authors did not present the explanation. The topographic differences affect the migration time of the knickpoints from low to high. For example, the adjustment time of river channel in Figure 5a1 is ~2 Ma, while that in Figure 5c1 is ~4 Ma, which are the conclusion of this manuscript. Moreover, from Figure 5a1 and c2, we can also see the difference in the channel slope, which is consistent with the difference in elevation. When the area and erosion coefficient remain basically unchanged, the slope differs by three times, but the erosion rate remains the same? This is not follow the rule of the stream power model ($E=KA^mS^n$). In Lines 234 and 236, the authors use the phrases of "wider divide" and "narrower divide". Therefore, a possible explanation is that the hillslope area is not included in the drainage area. However, the divide should be a curve without width. Therefore, I suggest that the authors check the setting conditions of the numerical simulation or give a reasonable explanation for this surprising phenomenon.

(2) The "transient slope change reversal" is an interesting phenomenon, and also the highlight of this manuscript. However, it is not difficult to understand that this phenomenon occurs in the numerical simulation under the set conditions. As the hillslope diffusion is not changed, the erosional difference between the river channel and the hillslope area will appear, when the rainfall changes. The erosion rate changes when the rainfall changes, while the hillslope erosion rate keeps. Then the "transient slope change reversal" phenomenon appears. Therefore, I admit that this phenomenon will occur in the numerical simulation of this study. However, in reality, the hillslope diffusion should be affected by the rainfall (Braun, 2018 GR). Therefore, Whether this phenomenon has practical significance has not been tested yet. At least the authors need to have some description on this in the Discussion section.

Other suggestions:
Figure 4: The blue profile should be 25 Ma, and the red profile should be 26 Ma.
Lines 192-199: This paragraph is exactly the same as the previous paragraph. Therefore, one of the paragraphs needs to be deleted.

---

## Author Comment (AC1)

**Response to comments from Anonymous Referee #1**

The manuscript is well-organized, clearly written, and presents a rigorous investigation into how landscapes respond to tectonic and climatic forcing, with a focus on slope evolution at river headwaters. Using the Badlands landscape evolution model, the authors demonstrate that changes in rainfall rate, unlike uplift rate changes, can induce transient slope change reversals near drainage divides, a process strongly modulated by hillslope diffusion. This insight offers a novel perspective on differentiating between climatic and tectonic controls in geomorphic systems. The experimental design is systematic, the interpretation is robust, and the study is presented in fluent and technically precise language. The work makes a valuable contribution to Earth surface process research and is suitable for publication with minor revisions. To further improve the clarity and scientific strength of the paper, the following suggestions are offered:

1. The authors argue that the transient slope change reversals are driven by differential erosion rates between the divide and adjacent areas. It is recommended to elaborate on why changes in rainfall rate lead to this contrast, while changes in uplift do not. What are the respective roles of hillslope diffusion and river incision in generating or amplifying these differences? A more detailed mechanistic discussion would strengthen the internal logic of the manuscript.

Thank you for this excellent suggestion. We agree that a more detailed mechanistic discussion was needed to strengthen the manuscript's logic. To address this, we have significantly expanded the discussion in the new **Section 4.1 Mechanism of transient slope change reversal**. The revised text now elaborates on why rainfall and uplift changes produce different responses by detailing the temporal mismatch between river incision and hillslope diffusion at the headwaters.

We have added the following text to Section 4.1:

The transient slope change reversal arises from a lag between two characteristic timescales: the hillslope response time and the channel incision response time. Following a change in rainfall, the river incision rate adjusts almost instantaneously. However, hillslope response lags behind channel response (Clubb et al., 2019). Hillslope diffusion, which controls sediment transport from divides to channels, is driven primarily by slope and remains initially unchanged. Near drainage divides, the river incision is weak and hillslope diffusion dominates (Dietrich et al., 2003). The temporal mismatch creates the observed imbalance at the headwaters. For instance, following a decrease in rainfall rate, sediment continues to diffuse toward the channel at pre-disturbance rates, but the ability of the channel to transport sediment is reduced due to lower discharge (Mitchell, 2020; Montgomery et al., 2000). This

imbalance causes the rate of sediment supply from hillslopes at the headwaters to exceed the rate of sediment removal by rivers, reducing channel slope temporarily and causing a transient slope change reversal. As the channel adjusts and the erosion wave migrates upstream, this reversal gradually disappears.

In contrast, a change in uplift rate uniformly raises the entire landscape without immediately affecting the efficiency of diffusion and incision. Because both the divide and its adjacent areas experience similar erosion conditions under constant discharge, no transient slope reversal occurs.

Notably, a lower Pe amplifies the imbalance between sediment supply from hillslopes and removal by rivers. This enlarges the zone where divide erosion rates differ from downstream areas. Therefore, the transient slope change reversal persists over a longer channel segment and for a longer duration, as observed in model M3 (Fig. 6 c2 and c3). In contrast, increasing Pe enhances river incision, which reduces the relative influence of diffusion. This leads to a shorter channel segment experiencing transient slope change reversal and a shorter duration of the transient response in model M4 (Fig. 6 d2 and d3).

2. Since the authors mention that slope-area and chi analysis can help identify transient slope change reversals, it is recommended to include representative plots from the numerical models. These visualizations would help illustrate how such plots reveal the geomorphic response to climatic or tectonic perturbations.

Thank you for this valuable suggestion. We have added a figure and the following text to Section 4.2:

In our models, a decrease in rainfall rate produces a localized flattening at high $\chi$ (headwaters), directly reflecting the transient slope-change reversal (Fig. 9). By contrast, in uplift-driven transients the $\chi$–elevation profile bows downward at low $\chi$, while the high-$\chi$ (headwater) segment remains straight and is simply translated upward. However, $\chi$–elevation analysis has limitations: it requires a steady-state baseline profile to distinguish different types of disturbances. $\chi$–elevation is therefore best used in concert with additional information, such as independent erosion-rate measurements, to robustly identify and attribute transient slope-change reversals.

[Figure]

**Figure 9. χ–elevation profiles of trunk streams in model M3 under three conditions: following an uplift rate increase (green), following a rainfall rate decrease (orange), and steady-state (light blue). The three gray dashed lines are parallel reference trends. χ–elevation profiles are calculated using a reference concavity index ($\theta_{ref}$) of 0.4.**

3. The manuscript suggests that erosion rate patterns near drainage divides, measurable through cosmogenic nuclides, may help distinguish between climatic and tectonic forcing. Including a brief geological example would help bridge the model findings with field-based applications and strengthen the practical relevance of the study.

Thank you for this great suggestion. We agree that a geological example would strengthen the practical relevance of our findings. Obtaining erosion patterns near drainage divides requires cosmogenic nuclide measurements at high spatial resolution across the divide zone. However, current studies focus on basin-averaged erosion rates with sparse sampling that cannot capture the spatial variation in erosion rates near divides that our model predicts. Our literature review confirmed that existing datasets lack the necessary spatial resolution to test our model predictions. This represents an important research gap that our study helps to identify. Perhaps, our paper will prompt experts to test our model.

4. In the methodology section, it is recommended that the authors clearly list all model parameters used, including the values of m and n in the stream power incision equation. Additionally, the initial condition is described as a uniform elevation of 10 meters; however, the landscape at the end of the first stage appears asymmetric. If any initial perturbation or

topographic noise was introduced to generate this asymmetry, it should be explicitly stated and justified in the methods section.

Thank you for this valuable suggestion. We have added a sentence stating that m = 0.5 and n = 1.0. We started with a perfectly flat 40 km × 80 km grid at 10 m elevation (400 m × 400 m cells). Badlands converts that grid into a triangular irregular network (TIN), which is asymmetric. Because erosion and diffusive transport in Badlands operate along those triangle edges (and depend on their length and slope), the tiny asymmetry of the TIN grows over time, yielding the slightly asymmetric topography at the end of Stage 1. We have verified that the asymmetry does not influence the conclusion.

5. Technical Corrections
Figure 4(a): The labels for 25 Ma and 26 Ma are reversed in the figure legend.

Thank you for catching this. We have corrected the time labels in the figure legend of Figure 4(a).

**References**

Clubb, F. J., Mudd, S. M., Hurst, M. D., & Grieve, S. W. D. (2019). Differences in channel and hillslope geometry record a migrating uplift wave at the Mendocino triple junction, California, USA. *Geology*, *48*(2), 184-188. https://doi.org/10.1130/g46939.1

Dietrich, W. E., Bellugi, D. G., Sklar, L. S., Stock, J. D., Heimsath, A. M., & Roering, J. J. (2003). Geomorphic Transport Laws for Predicting Landscape form and Dynamics. In *Prediction in Geomorphology* (pp. 103-132). https://doi.org/https://doi.org/10.1029/135GM09

Mitchell, S. B. (2020). Sediment transport and Marine Protected Areas. In *Marine Protected Areas* (pp. 587-598). https://doi.org/10.1016/b978-0-08-102698-4.00030-7

Montgomery, D. R., Zabowski, D., Ugolini, F. C., Hallberg, R. O., & Spaltenstein, H. (2000). 8 - Soils, Watershed Processes, and Marine Sediments. In M. C. Jacobson, R. J. Charlson, H. Rodhe, & G. H. Orians (Eds.), *International Geophysics* (Vol. 72, pp. 159-iv). Academic Press. https://doi.org/https://doi.org/10.1016/S0074-6142(00)80114-X

---

## Author Comment (AC2)

**Response to comments from Anonymous Referee #2**

The manuscript is well organized, and the topic is attractive. In the Introduction section, the authors made a good summary of the work of their predecessors, and then put forward their own new understanding through the method of numerical simulation. The method is reliable. I read the manuscript with great interest as it is full of important knowledge in the field of drainage system evolution. However, after careful reading and consideration, I think there are two significant issues which need to be addressed before the possible publication of this manuscript.

Major concerns:

(1) Why the mean elevation increase when the hillslope diffusion increase? This is counterintuitive. In Line 134, the authors also consider it as an astonishing phenomenon. However, the authors did not present the explanation. The topographic differences affect the migration time of the knickpoints from low to high. For example, the adjustment time of river channel in Figure 5a1 is ~2 Ma, while that in Figure 5c1 is ~4 Ma, which are the conclusion of this manuscript. Moreover, from Figure 5a1 and c2, we can also see the difference in the channel slope, which is consistent with the difference in elevation. When the area and erosion coefficient remain basically unchanged, the slope differs by three times, but the erosion rate remains the same? This is not follow the rule of the stream power model ($E=KA^mS^n$). In Lines 234 and 236, the authors use the phrases of "wider divide" and "narrower divide". Therefore, a possible explanation is that the hillslope area is not included in the drainage area. However, the divide should be a curve without width. Therefore, I suggest that the authors check the setting conditions of the numerical simulation or give a reasonable explanation for this surprising phenomenon.

Thank you for drawing attention to this point. To address this issue, we have calculated and added the "drainage density" (total channel length per unit drainage area) to Fig. 3 in the results. Drainage density decreases systematically as the hillslope diffusion coefficient increases. The runoff and erosion efficiency decreases as the drainage density decreases, resulting in an increase of landscape elevation.

[Figure]

**Fig. 3.** Hillshade maps showing erosion and deposition rates resulting from hillslope diffusion at the end of Stage 1 and the end of Stage 2 for models M1 (a), M2 (b), M3 (c), and M4 (d). Each model differs in hillslope diffusion coefficients ($k_{hl}$) and erodibility values ($k_d$). Blue areas indicate deposition, while red areas represent erosion. Color bar values indicate depositional (positive) and erosional (negative) rates (mm/yr). Numbers below each map display the mean elevation (black), drainage density (blue), and roughness (red). Dashed lines on maps at the end of Stage 1 denote the divides. The divides in Stage 2 are similar to those in Stage 1 and are not marked in this stage.

We have made some revision to Section 3.1. Before presenting the results of mean landscape elevation, drainage density, and surface roughness, we have added the following text:

To quantitatively compare landscape responses across our experiments, we compute three metrics: mean landscape elevation, drainage density, and surface roughness. Mean landscape elevation serves as an integrated measure of the overall erosional state of the landscape, representing the cumulative effect of tectonic uplift, channel incision, and hillslope processes on topographic development. Drainage density, defined as the ratio of total channel length to drainage basin area (Strahler, 1964), serves as a proxy for channel spacing and quantifies the degree of landscape dissection and runoff efficiency (Perron et al., 2008; Perron et al., 2009; Tassew et al., 2021). This metric provides insight into the spatial organization of the drainage network and its capacity to evacuate sediment and water from the landscape. Surface roughness quantifies the local topographic variability resulting from the

competing effects of processes that create and destroy relief (Doane et al., 2024). We calculate roughness as the difference between the maximum and minimum elevation values within a defined neighborhood surrounding each central pixel using the 'roughness' algorithm of GDAL in QGIS (Wilson et al., 2007).

We have changed *"Impact on patterns of drainage networks"* to *"Impact on drainage networks and density",* and added the following text to the end of this section:

However, when the erodibility remains constant, the drainage density decreases systematically with increasing diffusion coefficient in the order M1 > M2 > M3. This decrease in drainage density indicates wider valley spacing and reduced network tightness under stronger hillslope diffusion. M3 and M4 share the same hillslope diffusion coefficient, but the larger erodibility of M4 yields a higher drainage density than M3.

In addition, we have added the explanation to the end of Section 3.1:

Stronger diffusion smooths local slopes and reduces river incision rates under a constant uplift rate, while also widening valley spacing and lowering drainage density. Together, these effects have resulted in reduced drainage efficiency in some areas where the uplift rate exceeds the erosion rate, resulting in a higher mean elevation.

Finally, we agree with your suggestion that a divide should be a curve without width. Therefore, we have avoided using the phrases "wider divide" and "narrower divide" in the discussion.

(2) The"transient slope change reversal"is an interesting phenomenon, and also the highlight of this manuscript. However, it is not difficult to understand that this phenomenon occurs in the numerical simulation under the set conditions. As the hillslope diffusion is not changed, the erosional difference between the river channel and the hillslope area will appear, when the rainfall changes. The erosion rate changes when the rainfall changes, while the hillslope erosion rate keeps. Then the"transient slope change reversal"phenomenon appears. Therefore, I admit that this phenomenon will occur in the numerical simulation of this study. However, in reality, the hillslope diffusion should be affected by the rainfall (Braun, 2018 GR). Therefore, Whether this phenomenon has practical significance has not been tested yet. At least the authors need to have some description on this in the Discussion section.

Thank you for this insightful comment. We agree that the hillslope diffusion should be affected by the rainfall in nature. We have added a new section "**4.3 Model limitations**" to the discussion, acknowledging that the diffusion coefficient likely is not constant in response to climate change.

Section 4.3 is as follows:

In this study, we aim to explore the first-order impact of hillslope diffusion and river incision on landscape and consider a landscape evolving under the action of hillslope diffusion and river incision only. While the linear diffusion model is a common starting point, we acknowledge that it does not capture nonlinear processes, such as those driven by shallow landslides, which can become significant on steeper slopes (e.g., Jiménez-Hornero et al., 2005; Martin, 2000; Roering et al., 1999). Furthermore, our model does not account for potential feedback between climate and the diffusion coefficient itself. In natural settings, the hillslope diffusion coefficient can vary with climatic conditions via processes such as frost-crack weathering, and near-surface processes such as soil saturation, and root growth (Andersen et al., 2015; Bogaard & Greco, 2015; Braun, 2018; Gabet, 2000; Gabet & Mudd, 2010; Perron, 2017). Considering this feedback could introduce additional complexity. For instance, an increase in rainfall rate could increase the hillslope diffusion coefficient through higher soil moisture (Perron, 2017), potentially amplifying the transient slope change reversal. Conversely, a decrease in rainfall rate could decrease the hillslope diffusion coefficient and dampen the reversal. Future work could explore the parameter space where these feedbacks become significant.

In addition, our use of a detachment-limited stream power model simplifies the complexities of sediment flux. The "transient slope change reversal" we observe is fundamentally a result of a disequilibrium between hillslope sediment supply and the channel's transport capacity following a change in rainfall. A more complex model incorporating sediment transport dynamics (a "transport-limited" or "mixed" model) would likely modulate the magnitude and duration of this reversal.

Other suggestions:
Figure 4: The blue profile should be 25 Ma, and the red profile should be 26 Ma.

Thank you for catching this. We have corrected the time labels in the figure legend of Figure 4(a).

Lines 192-199: This paragraph is exactly the same as the previous paragraph. Therefore, one of the paragraphs needs to be deleted.

Thank you for catching this. We have deleted a repetitive paragraph.

**References**

Andersen, J. L., Egholm, D. L., Knudsen, M. F., Jansen, J. D., & Nielsen, S. B. (2015). The periglacial engine of mountain erosion – Part 1: Rates of frost cracking and frost creep. *Earth Surface Dynamics*, *3*(4), 447-462. https://doi.org/10.5194/esurf-3-447-2015

Bogaard, T. A., & Greco, R. (2015). Landslide hydrology: from hydrology to pore pressure. *WIREs Water*, *3*(3), 439-459. https://doi.org/10.1002/wat2.1126

Braun, J. (2018). A review of numerical modeling studies of passive margin escarpments leading to a new analytical expression for the rate of escarpment migration velocity. *Gondwana Research*, *53*, 209-224. https://doi.org/10.1016/j.gr.2017.04.012

Doane, T. H., Gearon, J. H., Martin, H. K., Yanites, B. J., & Edmonds, D. A. (2024). Topographic Roughness as an Emergent Property of Geomorphic Processes and Events. *AGU Advances*, *5*(5). https://doi.org/10.1029/2024av001264

Gabet, E. J. (2000). Gopher bioturbation: field evidence for non-linear hillslope diffusion. *Earth Surface Processes and Landforms*, *25*(13), 1419-1428.

Gabet, E. J., & Mudd, S. M. (2010). Bedrock erosion by root fracture and tree throw: A coupled biogeomorphic model to explore the humped soil production function and the persistence of hillslope soils. *Journal of Geophysical Research: Earth Surface*, *115*(F4). https://doi.org/10.1029/2009jf001526

Jiménez-Hornero, F. J., Laguna, A., & Giráldez, J. V. (2005). Evaluation of linear and nonlinear sediment transport equations using hillslope morphology. *Catena*, *64*(2-3), 272-280. https://doi.org/10.1016/j.catena.2005.09.001

Martin, Y. (2000). Modelling hillslope evolution: linear and nonlinear transport relations. *Geomorphology*, *34*(1-2), 1-21.

Perron, J. T. (2017). Climate and the Pace of Erosional Landscape Evolution. *Annual Review of Earth and Planetary Sciences*, *45*(1), 561-591. https://doi.org/10.1146/annurev-earth-060614-105405

Perron, J. T., Dietrich, W. E., & Kirchner, J. W. (2008). Controls on the spacing of first-order valleys. *Journal of Geophysical Research*, *113*(F4). https://doi.org/10.1029/2007jf000977

Perron, J. T., Kirchner, J. W., & Dietrich, W. E. (2009). Formation of evenly spaced ridges and valleys. *Nature*, *460*(7254), 502-505. https://doi.org/10.1038/nature08174

Roering, J. J., Kirchner, J. W., & Dietrich, W. E. (1999). Evidence for nonlinear, diffusive sediment transport on hillslopes and implications for landscape morphology. *Water Resources Research*, *35*(3), 853-870. https://doi.org/10.1029/1998wr900090

Strahler, A. N. (1964). Quantitative geomorphology of drainage basin and channel networks. *Handbook of applied hydrology*.

Tassew, B. G., Belete, M. A., & Miegel, K. (2021). Assessment and analysis of morphometric characteristics of Lake Tana sub-basin, Upper Blue Nile Basin, Ethiopia. *International Journal of River Basin Management*, *21*(2), 195-209. https://doi.org/10.1080/15715124.2021.1938091

Wilson, M. F. J., O'Connell, B., Brown, C., Guinan, J. C., & Grehan, A. J. (2007). Multiscale Terrain Analysis of Multibeam Bathymetry Data for Habitat Mapping on the Continental Slope. *Marine Geodesy*, *30*(1-2), 3-35. https://doi.org/10.1080/01490410701295962

---

## Author Comment (AC3)

**Response to comments from Anonymous Referee #3**

Summary

In this contribution the authors aim to examine how topography responds to changes in uplift versus climate using the streampower+diffusion landscape evolution model. They specifically focus on how diffusion modulates the response of channel profiles to changes in rainfall rate in the streampower model. They find a nonmonotonic response of channel slope near ridges when precipitation is increased or decreased, which doesn't appear in the absence of diffusion, or when the uplift rate is changed. They propose that this occurs because diffusion-driven erosion is unaffected by change in precipitation, inducing a local change in the balance of advection versus diffusion processes near channel heads. They suggest that this could be a diagnostic feature of response to changes in climate.

While this is an interesting signature that I don't think has been described before, the authors have missed a few important points that call into question the usefulness of the feature they have described. First, and most importantly, their method relies on the idea that the diffusion term is insensitive to changes in climate. This seems highly unlikely, given that slope stability is sensitive to hydrological processes (Bogaard & Greco, 2016), and other biophysical processes that drive soil production and creep are almost certainly climate-sensitive (Andersen et al., 2015; Gabet, 2000; Gabet & Mudd, 2010). While theories that clearly link hillslope processes to climate are still needed, it is generally accepted that both soil production and the diffusion coefficient increase with mean annual precipitation (Perron, 2017).

Thank you for this insightful comment. We agree that our model's assumption of a climate-insensitive diffusion coefficient is a critical simplification. In this study, we intentionally hold the diffusion coefficient constant to isolate how a given level of diffusive efficiency modulates the transient channel slope response to step changes in uplift or rainfall rate. This simplification allows us to identify the baseline "transient slope change reversal" mechanism, which could be obscured by more complex, competing feedbacks. We have added a new section "**4.3 Model limitations**" to the discussion, acknowledging that the diffusion coefficient likely is not constant in response to climate change.

Section 4.3 is as follows:

In this study, we aim to explore the first-order impact of hillslope diffusion and river incision on landscape and consider a landscape evolving under the action of hillslope diffusion and river incision only. While the linear diffusion model is a common starting point, we acknowledge that it does not capture nonlinear processes, such as those driven by shallow landslides, which can become significant on steeper slopes (e.g., Jiménez-Hornero et al., 2005; Martin, 2000; Roering et al., 1999). Furthermore, our model does not account for potential feedback between climate and the diffusion coefficient itself.

In natural settings, the hillslope diffusion coefficient can vary with climatic conditions via processes such as frost-crack weathering, and near-surface processes such as soil saturation, and root growth (Andersen et al., 2015; Bogaard & Greco, 2015; Braun, 2018; Gabet, 2000; Gabet & Mudd, 2010; Perron, 2017). Considering this feedback could introduce additional complexity. For instance, an increase in rainfall rate could increase the hillslope diffusion coefficient through higher soil moisture (Perron, 2017), potentially amplifying the transient slope change reversal. Conversely, a decrease in rainfall rate could decrease the hillslope diffusion coefficient and dampen the reversal. Future work could explore the parameter space where these feedbacks become significant.

In addition, our use of a detachment-limited stream power model simplifies the complexities of sediment flux. The "transient slope change reversal" we observe is fundamentally a result of a disequilibrium between hillslope sediment supply and the channel's transport capacity following a change in rainfall. A more complex model incorporating sediment transport dynamics (a "transport-limited" or "mixed" model) would likely modulate the magnitude and duration of this reversal.

A second related issue is distilling the effects of climate change down to a linear increase in average precipitation. Climate change is manifested in changes to the not just the mean, but also the distribution of event magnitudes and the phase (snow, rain) of precipitation due to changing temperature, which will be especially important in mountainous settings such as those considered (Meira Neto et al., 2020). Settings respond to precipitation changes differently depending on the dominant runoff generation mechanisms (Uhlenbrook et al., 2005), which are further modulated by erosion thresholds (DiBiase & Whipple, 2011). Such thresholds are especially important in headwaters, where the authors report their slope effect. Furthermore, geomorphic models that consider vegetation response to climate change suggest that the erosion response to precipitation change could even be reversed due to dynamic feedbacks with vegetation cover and evapotranspiration (Yetemen et al., 2019). None of these processes are mentioned in the present paper.

Thank you for this detailed and constructive feedback. You have raised several excellent points about the real-world complexities of climate change that our simplified model does not capture.

Based on this valuable feedback, we agree that using the term "Climate" was too broad for our study's scope. We have revised the title to be more precise: "**Rainfall and Tectonic Forcing Lead to Contrasting Headwater Slope Evolutions**". As you have noted, our study simplifies the climate forcing to a change in the mean rainfall rate. This simplified approach allows us to isolate the outcome of fundamental processes that would be obscured in more complex and realistic scenarios.

In addition, we have added the following text after the introduction of Equation 1 in Section 1:

Eq. (1) simplifies the impact of climate on erosion. However, real landscapes respond to climate change through shifts not only in mean $P$ but also in (i) the distribution of storm magnitudes, (ii) the phase of precipitation (snow vs. rain) that controls the timing of snowmelt runoff (Meira Neto et al., 2020), and (iii) the dominant runoff-generation mechanism (Uhlenbrook et al., 2005). Moreover, incision in channels is often controlled by erosion thresholds (DiBiase & Whipple, 2011) and may be further moderated by vegetation–evapotranspiration feedbacks (Yetemen et al., 2019). While these factors are critical for site-specific predictions, Eq. (1) is used here to isolate the first-order impact of a change in fluvial erosion efficiency on landscape form, providing a baseline for understanding these more complex interactions.

While I'm unsure that the streampower+diffusion model is the right tool to answer questions of climate sensitivity, I understand the tendency to stick with it in the name of interpretable simplicity. One of the reasons to stick with this model is because of its well-developed nondimensional forms (Bonetti et al., 2020; Litwin et al., 2025; Perron et al., 2008; Theodoratos et al., 2018), which provide clear methods for understanding fundamental process competition. The authors run into the problem of non-uniqueness in process competition when they change the streampower coefficient and diffusion coefficient but maintain their ratio. However, they do not provide any explanation of the fundamental scaling between the two, which is well understood (e.g., Perron et al., 2008).

Thank you for emphasizing the value of established nondimensional forms. We have clarified our methodology by explicitly computing the dimensionless Pe number, a well-established measure of the relative dominance of advective (fluvial) vs. diffusive processes (Perron et al., 2008, 2009). While varying streampower coefficient ($k_d$) and diffusion coefficient ($k_{hl}$) while holding Pe constant could theoretically produce similar steady-state forms, we emphasize that our analysis focuses on transient dynamics, where the absolute values of $k_d$ and $k_{hl}$ (not just their ratio) could influence response times. By adopting the dimensionless framework, we align our work with fundamental scaling theory (e.g., Perron et al., 2008) while retaining the interpretability of our results.

We have added the following text to Section 2:

For each model, we compute the dimensionless parameter Pe to combine two a priori independent parameters (the diffusion coefficient $k_{hl}$ and the erodibility $k_d$) into a single dimensionless measure of process competition (Perron et al., 2008; Perron et al., 2009):

$$\text{Pe} \ = \frac{k_d \, l^{2m+1}}{k_{hl}} \tag{4}$$

Pe is analogous to a Péclet number, which is the ratio of a diffusion timescale to an advection timescale (Perron et al., 2008). Low Pe values indicate diffusion-dominated systems, while high values indicate advection-dominated systems. We take the characteristic horizontal length scale $l$ to be 40 km, representative of the real landscape. Based on our parameter values, model M3 has the lowest Pe, indicating that diffusion is more dominant in this model than in the others. Furthermore, models M2 and M4 share the same Pe because their parameters for $k_{hl}$ and $k_d$ were both doubled in M4 relative to M2, keeping their ratio constant.

**Table 1. Diffusion coefficient and erodibility of four models**

| Model | Diffusion coefficient $k_{hl}$ (m²/yr) | Erodibility $k_d$ (1/yr) | Pe |
|-------|----------------------------------------|--------------------------|-----|
| M1 | 0 | $2.3 \times 10^{-6}$ | ∞ (no diffusion) |
| M2 | 1 | $2.3 \times 10^{-6}$ | 3680 |
| M3 | 2 | $2.3 \times 10^{-6}$ | 1840 |
| M4 | 2 | $4.6 \times 10^{-6}$ | 3680 |

Overall, I think this paper needs substantial work to become a valuable contribution. My main recommendation would be to engage with models that link changes in climatic, hydrological, and geomorphic processes in some more realistic level of detail. If not, they could describe the deficiencies of the streampower+diffusion model and provide a more comprehensive description of the effect they observe, using available nondimensional frameworks and acknowledging that the diffusion coefficient likely is not constant in response to climate change.

Thank you for this insightful comment and great suggestion. We have made clear in the revised manuscript that as in many other modelling-based research, our intention is not to simulate nature in all its complexity, but to understand the role of individual processes. According to your advice, we have added a new section "**4.3 Model limitations**" in the discussion to describe the deficiencies of the streampower+diffusion model and acknowledged that the diffusion coefficient likely is not constant in response to climate change. In addition, we have used the nondimensional Pe number to provide a better description of our results.

Line-by-line comments

1.  Transient slope change reversals are not yet defined.

Thank you for this comment. We have defined transient slope change reversals in Section 3.2.2 as follows:

In contrast, a decrease in rainfall rate triggers a "*transient slope change reversal*", a phenomenon we define as a non-monotonic adjustment where the headwater channel slope initially changes in the opposite direction of its final steady state.

2.  Needs more description of how changes in climate actually yield changes in runoff production. Q=PA is a little too simple.

Thank you for this valuable comment. We have added the following text following the introduction of Equation 1 in Section 1:

Eq. (1) simplifies the impact of climate on erosion. However, real landscapes respond to climate change through shifts not only in mean *P* but also in (i) the distribution of storm magnitudes, (ii) the phase of precipitation (snow vs. rain) that controls the timing of snowmelt runoff (Meira Neto et al., 2020), and (iii) the dominant runoff-generation mechanism (Uhlenbrook et al., 2005). Moreover, incision in channels is often controlled by erosion thresholds (DiBiase & Whipple, 2011) and may be further moderated by vegetation–evapotranspiration feedbacks (Yetemen et al., 2019). While these factors are critical for site-specific predictions, Eq. (1) is used here to isolate the first-order impact of a change in fluvial erosion efficiency on landscape form, providing a baseline for understanding these more complex interactions.

48-49. Needs better description of what causes diffusion processes, how they might be linked to climate as well.

Thank you for pointing it out. We have expanded the relevant paragraph in Section 1. The revised text now provides a more detailed description of the physical processes represented by the diffusion coefficient, such as soil creep, rain splash, and bioturbation. Furthermore, it now explicitly describes how these processes are linked to climate through factors like temperature, moisture, and vegetation cover.

We have added the following text in Section 1:

Hillslope diffusion is the result of a combination of multiple near-surface processes: (i) rainsplash and sheet-flow creep driven by raindrop impact and overland flow (Guy et al., 1987; Meyer et al., 1975;

Young & Wiersma, 1973), (ii) soil creep produced by cyclical wetting-drying, shrink–swell, and freeze–thaw strains (Anderson & Anderson, 2010), (iii) bioturbation by burrowing animals and tree throw that mix and move regolith (Gabet et al., 2003; Roering et al., 2010), and (iv) small shallow landslides that act diffusively when averaged over long timescales (Martin, 2000).

Climate controls the relative efficiency of these mechanisms. Mean annual precipitation and storm magnitudes regulate rainsplash fluxes and influence vegetation density, which in turn affects soil creep (Istanbulluoglu & Bras, 2006). Freeze–thaw frequency, governed by temperature and moisture, dictates the rate of frost creep and solifluction in high-altitude or high-latitude settings (Hales & Roering, 2007).

62-63. Part of the reason this hasn't been explored is because we don't have adequate theory describing how the diffusion coefficient changes with climate, although it almost certainly does.

Thank you for this very insightful comment. We agree completely. To incorporate this important context into our paper, we have added a sentence to our Introduction that acknowledges this theoretical gap as a reason for the problem being under-studied. We believe this strengthens the rationale for our work.

The sentence is as follows:

This knowledge gap exists in part because there is not yet a comprehensive theory describing how the hillslope diffusion coefficient changes with climate.

84-86. Needs citation.

Thank you. We have added the citation.

91-94. Maybe just describe the processes that are relevant. No marine? Source to sink?

Thank you for this great suggestion. We have deleted irrelevant processes.

Table 1. Those are really large values of the diffusion coefficient! Usually, find values on the order of 0.001-0.01 m2/yr using hilltop curvature and erosion rates. The sensitivity to the value used is dependent on the grid size, and there are already well-established nondimensional forms that can help describe this (Bonetti et al., 2020; Litwin et al., 2025). It might be useful to consider those.

1. Again, this would be evident if you used established nondimensionalizations.

Thank you for this comment. We agree that the absolute values for the diffusion coefficient used in our models are larger than those typically measured in the field. We have used the established Pe number from Perron et al. (2008). The Pe values used in our models range

from 1840 to 3680 for models M2-M4, which are comparable to those explored in the foundational work of Perron et al. (2008). This suggests that our choices of diffusion coefficients are consistent with the widely accepted ranges of Pe and are appropriate for our study context.

2. You have not explained why topographic roughness is a useful or interesting metric, or how you are calculating it.

Thank you for this comment. Topographic roughness is a crucial metric because it quantifies the local variability in surface relief caused by incision and diffusion processes, which are central to landscape evolution. Understanding roughness helps us investigate how the landscape responds to perturbations like changes in uplift or rainfall and provides insight into the spatial variability across different models. We calculate roughness using the GDAL 'roughness' algorithm in QGIS (Wilson et al., 2007), which measures the difference between the maximum and minimum elevation values within a defined neighborhood around each central pixel.

We have added the following text in Section 3.1:

To quantitatively compare landscape responses across our experiments, we compute three metrics: mean landscape elevation, drainage density, and surface roughness. Mean landscape elevation serves as an integrated measure of the overall erosional state of the landscape, representing the cumulative effect of tectonic uplift, channel incision, and hillslope processes on topographic development. Drainage density, defined as the ratio of total channel length to drainage basin area (Strahler, 1964), serves as a proxy for channel spacing and quantifies the degree of landscape dissection and runoff efficiency (Perron et al., 2008; Perron et al., 2009; Tassew et al., 2021). This metric provides insight into the spatial organization of the drainage network and its capacity to evacuate sediment and water from the landscape. Surface roughness quantifies the local topographic variability resulting from the competing effects of processes that create and destroy relief (Doane et al., 2024). We calculate roughness as the difference between the maximum and minimum elevation values within a defined neighborhood surrounding each central pixel using the 'roughness' algorithm of GDAL in QGIS (Wilson et al., 2007).

3.2 "rivers' channel"

Fig. 4 Here it sounds like the effect of diffusion is unimportant, but in subsequent figures, it clearly is important. You could just explain Figure 4 as the kind of null case.

Thank you for this great suggestion. We have divided Section 3.2 into these two subsections: **3.2.1 Null-case control (Model M1, $k_{hl} = 0$)** and **3.2.2 Diffusion-enabled models (M2–M4)** to present the results more clearly and logically.

1.  "Monotonously"

2.  Needs to be more specific.

Thank you for this comment. We have revised the text and replaced "Monotonously" with "monotonically" in Section 3.2.1. The revised text is as follows:

Notably, within 1-2 Myrs of the change in rainfall or uplift rates, the channel elevation at the headwaters changes, but the slope remains nearly constant (Fig. 5 a1-3 and Fig. 6 a1-3). As the erosion wave approaches the headwaters, the channel slope increases or decreases monotonically and eventually stabilizes.

190-191. Not all channels experience this effect? Is there a threshold where it starts to occur?

Thank you for this comment. Transient reversal occurs in all channels. The extent of reversal becomes larger when Pe is lower. No threshold exists.

We have revised the relevant text in Section 3.2.2:

We do not find a distinct threshold for the initiation of the transient slope change reversal; rather, it is present whenever hillslope diffusion is active (Pe < ∞). The primary control on the reversal is its magnitude and persistence, which vary continuously with Pe. Our results show that landscapes with lower Pe values, where hillslope diffusion is more dominant relative to channel incision, exhibit more pronounced and persistent reversals. For example, model M3, which has the lowest Pe, shows a reversal that persists longer and extends over a longer channel segment compared to other models (Fig. 6 c1-3).

192-199. Just a copy of the previous text.

Thank you for catching this. We have deleted a repetitive paragraph.

**References**

Andersen, J. L., Egholm, D. L., Knudsen, M. F., Jansen, J. D., & Nielsen, S. B. (2015). The periglacial engine of mountain erosion – Part 1: Rates of frost cracking and frost creep. *Earth Surface Dynamics*, *3*(4), 447-462. https://doi.org/10.5194/esurf-3-447-2015

Anderson, R. S., & Anderson, S. P. (2010). *Geomorphology: the mechanics and chemistry of landscapes*. Cambridge University Press.

Bogaard, T. A., & Greco, R. (2015). Landslide hydrology: from hydrology to pore pressure. *WIREs Water*, *3*(3), 439-459. https://doi.org/10.1002/wat2.1126

Braun, J. (2018). A review of numerical modeling studies of passive margin escarpments leading to a new analytical expression for the rate of escarpment migration velocity. *Gondwana Research*, *53*, 209-224. https://doi.org/10.1016/j.gr.2017.04.012

DiBiase, R. A., & Whipple, K. X. (2011). The influence of erosion thresholds and runoff variability on the relationships among topography, climate, and erosion rate. *Journal of Geophysical Research*, *116*(F4). https://doi.org/10.1029/2011jf002095

Doane, T. H., Gearon, J. H., Martin, H. K., Yanites, B. J., & Edmonds, D. A. (2024). Topographic Roughness as an Emergent Property of Geomorphic Processes and Events. *AGU Advances*, *5*(5). https://doi.org/10.1029/2024av001264

Gabet, E. J. (2000). Gopher bioturbation: field evidence for non-linear hillslope diffusion. *Earth Surface Processes and Landforms*, *25*(13), 1419-1428.

Gabet, E. J., & Mudd, S. M. (2010). Bedrock erosion by root fracture and tree throw: A coupled biogeomorphic model to explore the humped soil production function and the persistence of hillslope soils. *Journal of Geophysical Research: Earth Surface*, *115*(F4). https://doi.org/10.1029/2009jf001526

Gabet, E. J., Reichman, O. J., & Seabloom, E. W. (2003). The Effects of Bioturbation on Soil Processes and Sediment Transport. *Annual Review of Earth and Planetary Sciences*, *31*(1), 249-273. https://doi.org/10.1146/annurev.earth.31.100901.141314

Guy, B., Dickinson, W., & Rudra, R. (1987). The roles of rainfall and runoff in the sediment transport capacity of interrill flow. *Transactions of the ASAE*, *30*(5), 1378-1386.

Hales, T. C., & Roering, J. J. (2007). Climatic controls on frost cracking and implications for the evolution of bedrock landscapes. *Journal of Geophysical Research: Earth Surface*, *112*(F2). https://doi.org/10.1029/2006jf000616

Istanbulluoglu, E., & Bras, R. L. (2006). On the dynamics of soil moisture, vegetation, and erosion: Implications of climate variability and change. *Water Resources Research*, *42*(6). https://doi.org/10.1029/2005wr004113

Jiménez-Hornero, F. J., Laguna, A., & Giráldez, J. V. (2005). Evaluation of linear and nonlinear sediment transport equations using hillslope morphology. *Catena*, *64*(2-3), 272-280. https://doi.org/10.1016/j.catena.2005.09.001

Martin, Y. (2000). Modelling hillslope evolution: linear and nonlinear transport relations. *Geomorphology*, *34*(1-2), 1-21.

Meira Neto, A. A., Niu, G.-Y., Roy, T., Tyler, S., & Troch, P. A. (2020). Interactions between snow cover and evaporation lead to higher sensitivity of streamflow to temperature. *Communications Earth & Environment*, *1*(1). https://doi.org/10.1038/s43247-020-00056-9

Meyer, L. D., Foster, G. R., & Römkens, M. J. M. (1975). Source of soil eroded by water from upland slopes. In *Present and prospective technology for predicting sediment yields and sources* (pp. 177-189). USDA-ARS, U.S. Gov. Print. Office.

Perron, J. T. (2017). Climate and the Pace of Erosional Landscape Evolution. *Annual Review of Earth and Planetary Sciences*, *45*(1), 561-591. https://doi.org/10.1146/annurev-earth-060614-105405

Perron, J. T., Dietrich, W. E., & Kirchner, J. W. (2008). Controls on the spacing of first-order valleys. *Journal of Geophysical Research*, *113*(F4). https://doi.org/10.1029/2007jf000977

Perron, J. T., Kirchner, J. W., & Dietrich, W. E. (2009). Formation of evenly spaced ridges and valleys. *Nature*, *460*(7254), 502-505. https://doi.org/10.1038/nature08174

Roering, J. J., Kirchner, J. W., & Dietrich, W. E. (1999). Evidence for nonlinear, diffusive sediment transport on hillslopes and implications for landscape morphology. *Water Resources Research*, *35*(3), 853-870. https://doi.org/10.1029/1998wr900090

Roering, J. J., Marshall, J., Booth, A. M., Mort, M., & Jin, Q. (2010). Evidence for biotic controls on topography and soil production. *Earth and Planetary Science Letters*, *298*(1-2), 183-190. https://doi.org/10.1016/j.epsl.2010.07.040

Strahler, A. N. (1964). Quantitative geomorphology of drainage basin and channel networks. *Handbook of applied hydrology*.

Tassew, B. G., Belete, M. A., & Miegel, K. (2021). Assessment and analysis of morphometric characteristics of Lake Tana sub-basin, Upper Blue Nile Basin, Ethiopia. *International Journal of River Basin Management*, *21*(2), 195-209. https://doi.org/10.1080/15715124.2021.1938091

Uhlenbrook, S., Didszun, J., & Leibundgut, C. (2005). Runoff generation processes on hillslopes and their susceptibility to global change. *Global Change and Mountain Regions: An Overview of Current Knowledge*, 297-307.

Wilson, M. F. J., O'Connell, B., Brown, C., Guinan, J. C., & Grehan, A. J. (2007). Multiscale Terrain Analysis of Multibeam Bathymetry Data for Habitat Mapping on the Continental Slope. *Marine Geodesy*, *30*(1-2), 3-35. https://doi.org/10.1080/01490410701295962

Yetemen, O., Saco, P. M., & Istanbulluoglu, E. (2019). Ecohydrology controls the geomorphic response to climate change. *Geophysical Research Letters*, *46*(15), 8852-8861.

Young, R. A., & Wiersma, J. (1973). The role of rainfall impact in soil detachment and transport. *Water Resources Research*, *9*(6), 1629-1636.

---

## Editor Decision (ED1)

[revised manuscript text omitted]

$$\frac{\partial z}{\partial t} = k_{hl} \, \nabla^2 z \tag{3}$$

where $k_{hl}$ is the hillslope diffusion coefficient, which integrates climate, lithology, soil conditions, and biotic influences (Dietrich and Perron, 2006; Hurst et al., 2013; Robl et al., 2017). Hillslope diffusion is the result of a combination of multiple near-surface processes: (i) rainsplash and sheet-flow creep driven by raindrop impact and overland flow (Guy et al., 1987; Meyer et al., 1975; Young and Wiersma, 1973), (ii) soil creep produced by cyclical wetting-drying, shrink–swell, and freeze–thaw strains (Anderson & Anderson, 2010), (iii) bioturbation by burrowing animals and tree throw that mix and move regolith (Gabet, 2003; Roering et al., 2010), and (iv) small shallow landslides that act diffusively when averaged over long timescales (Martin, 2000).

Climate controls the relative efficiency of these mechanisms. Mean annual precipitation and storm magnitudes regulate rainsplash fluxes and influence vegetation density, which in turn affects soil creep (Istanbulluoglu and Bras, 2006). Freeze–thaw frequency, governed by temperature and moisture, dictates the rate of frost creep and solifluction in high-altitude or high-latitude settings (Hales & Roering, 2007). Hillslope diffusion gradually transports soil and sediment downslope due to gravity and reshapes substantially the landscape over time (e.g., Litwin et al., 2025; Perron et al., 2008; Roering, 2008). It has been shown that hillslope diffusion strongly influences drainage density and valley spacing (Perron et al., 2008; Sweeney et al., 2015; Tucker and Bras, 1998). Additionally, the sediment and soil transported from hillslopes impact river incision by either acting as tools for erosion or forming a protective cover that shields the underlying bedrock from further erosion (Sklar and Dietrich, 2001).

While much research has focused on river channel evolution (e.g., Kirby and Whipple, 2012; Wobus et al., 2010), few have explored whether and how river channels respond differently to tectonic and climatic changes when hillslope diffusion is included. This knowledge gap exists in part because there is not yet a comprehensive theory describing how the hillslope diffusion coefficient changes with climate. Before addressing this issue, the following paragraph clarifies the notions of steady-state and transient landscapes.

**1.2 Steady state vs transient landscapes**

Computer-generated landscapes evolving under controlled tectonic and climatic conditions provide a robust framework for better understanding the formation and evolution of natural landscapes (e.g., Chen et al., 2014; Pan et al., 2021; Salles and Hardiman, 2016; Schwanghart and Scherler, 2014). These models show that a landscape reaches a steady state when the uplift rate equals the erosion rate. When the uplift rate changes, landscapes are in a transient state of disequilibrium and evolve to reach a new steady state (e.g., Leonard and Whipple, 2021; Miller et al., 2012; O'hara et al., 2019). Steady-state and transient landscapes show a sharp contrast in the morphology of river profiles. When a river channel has reached a steady state, its longitudinal elevation profile is usually smooth and concave-up (Fig. 1a). In contrast, under uniform lithology, knickpoints form in transient river channels (Wobus et al., 2006b; Lague, 2014; Neely et al., 2017; Whipple et al., 2013). A knickpoint is a location where there is an abrupt change in the channel slope (Fig. 1b). A positive knickpoint forms where the slope suddenly increases downstream, while a negative knickpoint forms where the slope decreases abruptly. A mobile positive knickpoint indicates an increase in uplift rate and/or a decrease in erosion efficiency (induced by a decrease in rainfall rate, for example),

95   while a mobile negative knickpoint indicates the opposite conditions (Baldwin et al., 2003). Both types of knickpoints typically form at the river mouth and migrate upstream toward the headwaters.

[Figure]

**Figure 1. Channel profiles with different morphology. (a) a steady-state river profile. (b) Transient river profiles with a negative or**
100   **positive slope-break knickpoint.**

A migrating knickpoint separates the channel into two segments, upstream and downstream segments. It has been proposed that regardless of whether the transient change is driven by tectonics or climate, the elevation of the upstream segment changes while its slope remains constant (Whipple, 2001). After the downstream segment reaches a steady state, its channel elevation
105   and slope have changed (e.g., Whipple, 2001; Whipple and Tucker, 1999).

**2 Methodology and model setup**

To investigate landscape evolution under climatic or tectonic changes, as well as varying erodibility and hillslope diffusion, we use the long-term surface evolution model Badlands (Basin and Landscape Dynamics) (Salles, 2016; Salles and Hardiman, 2016). Badlands can be used to simulate landscape development via the mobilisation of sediments through hillslope diffusion
110   and stream-power incision. Our model assumes that hillslope sediment transport rates are linearly proportional to the slope gradient. Here, we explore landscape responses to changes in rainfall or uplift, and we disregard isostatic re-adjustment. In particular, we focus on contrasts in drainage network patterns, average elevation, surface roughness, and river profiles. Our initial landscape models are mapped over a 40 km × 80 km grid with a uniform initial elevation of 10 m and a spatial resolution of 400 m × 400 m. We design four initial models with varying hillslope diffusion and erodibility coefficients (Table
115   1). The diffusion coefficient is set to 0 in model M1, meaning the landscape evolution is purely driven by riverine processes with an erodibility coefficient of $2.3 \times 10^{-6}$ yr$^{-1}$. We set the diffusion coefficient to 1 m$^2$/yr in model M2 and 2 m$^2$/yr in model M3. Finally, in our last model M4, the erodibility is doubled to $4.6 \times 10^{-6}$ yr$^{-1}$. In all cases, the stream-power law uses m = 0.5 and n = 1.0.

For each model, we compute the dimensionless parameter Pe to combine two a priori independent parameters (the diffusion
120  coefficient $k_{hl}$ and the erodibility $k_d$) into a single dimensionless measure of process competition (Bonetti et al., 2020; Perron
et al., 2008; Perron et al., 2009):

$$\text{Pe} = \frac{k_d P^m l^{2m+1}}{k_{hl}} \tag{4}$$

Pe is analogous to a Péclet number, which is the ratio of a diffusion timescale to an advection timescale (Perron et al., 2008).
Low Pe values indicate diffusion-dominated systems, while high values indicate advection-dominated systems. We take the
125  characteristic horizontal length scale $l$ to be 40 km, representative of the real landscape. Based on our parameter values, model
M3 has the lowest Pe, indicating that diffusion is more dominant in this model than in the others. Furthermore, models M2
and M4 share the same Pe because their parameters for $k_{hl}$ and $k_d$ are both doubled in M4 relative to M2, keeping their ratio
constant.

**Table 1. Diffusion coefficient, erodibility, and initial Pe of four models**

| Model | Diffusion coefficient $k_{hl}$ (m$^2$/yr) | Erodibility $k_d$ (1/yr) | Initial Pe (Rainfall = 2 m/yr) |
|---|---|---|---|
| M1 | 0 | $2.3 \times 10^{-6}$ | ∞ (no diffusion) |
| M2 | 1 | $2.3 \times 10^{-6}$ | 5204 |
| M3 | 2 | $2.3 \times 10^{-6}$ | 2602 |
| M4 | 2 | $4.6 \times 10^{-6}$ | 5204 |

130

Our four models are submitted to a combination of uniform uplift at a rate of 300 m/Myr and background rainfall at a rate of
2 m/yr until they reach a steady-state equilibrium, where mean elevation and river profiles no longer change (Montgomery,
2001; Willett & Brandon, 2002). This first stage lasts for 25 Myr (Fig. 2), after which all models reach a steady state.
In the second stage, which also lasts 25 Myr, each model is subjected to a perturbation while the other forcing remains constant.
135  We either:

- Increase rainfall to 6 m/yr or decrease it to 0.67 m/yr, while keeping uplift fixed at 300 m/Myr, or
- Increase uplift to 900 m/Myr or decrease it to 100 m/Myr, while keeping rainfall fixed at 2 m/yr.

[Figure]

**Figure 2. Each of our four initial models (M1 to M4) experiences four different two-stage landscape evolutions controlled by changes in rainfall or uplift. Stage 1: An initial flat landscape is uplifted under an uplift rate of 300 m/Myr and a rainfall rate of 2 m/yr until a steady-state landscape is reached. Stage 2: Changes in rainfall or uplift rate.**

This design yields 16 individual experiments (Fig. 2), allowing us to assess landscape responses to changes in rainfall and uplift rates separately. A key consequence of our experimental design is that a change in rainfall rate directly changes the advection timescale associated with river incision. Thus, this changes Pe and alters the fundamental balance between advective and diffusive processes, as shown in Stage 2 (Table 2).

**Table 2. Pe for Stage 2 rainfall-change scenarios**

| Model | Pe (Rainfall = 0.67 m/yr) | Pe (Rainfall = 6 m/yr) |
|-------|---------------------------|------------------------|
| M2 | 3004 | 9014 |
| M3 | 1502 | 4507 |
| M4 | 3004 | 9014 |

**3 Results**

**3.1 Comparison of final, steady-state landscapes**

To quantitatively compare landscape responses across our experiments, we compute three metrics: mean landscape elevation, drainage density, and surface roughness. Mean landscape elevation serves as an integrated measure of the overall erosional state of the landscape, reflecting the cumulative effect of tectonic uplift, channel incision, and hillslope processes on

155   topographic development. Drainage density, defined as the ratio of total channel length to drainage basin area (Strahler, 1964), acts as a proxy for channel spacing and quantifies the degree of landscape dissection and runoff efficiency (Tassew et al., 2021; Perron et al., 2009; Perron et al., 2008). This metric provides insight into the spatial organization of the drainage network and its capacity to evacuate sediment and water from the landscape. Surface roughness quantifies the local topographic variability resulting from the competing effects of processes that create and destroy relief (Doane et al., 2024). We calculate roughness

160   as the difference between the maximum and minimum elevation values within a defined neighborhood surrounding each central pixel using the 'roughness' algorithm of GDAL in QGIS (Wilson et al., 2007).

*Impact on drainage networks and density*: Despite having different erodibility and diffusion coefficients and going through different climatic and tectonic histories, our four initial models display broadly similar patterns of drainage networks. In all 16 cases, the two largest drainage basins form at the eastern and western parts of the landscape, separated by a central divide (Fig.

165   3). The drainage patterns in models M2 and M4 are highly similar, reflecting that both models have the same Pe value. However, when the erodibility remains constant, the drainage density decreases systematically with increasing diffusion coefficient in the order M1 > M2 > M3. This decrease in drainage density indicates wider valley spacing and reduced network tightness under stronger hillslope diffusion. M3 and M4 share the same hillslope diffusion coefficient, but the larger erodibility of M4 yields a higher drainage density than M3.

170   *Impact on average elevation and surface roughness*: Our results show that the mean landscape elevation and surface roughness increase following a decrease in rainfall rate or an increase in uplift rate, and decrease following an increase in rainfall rate or a decrease in uplift rate. Regardless of rainfall or uplift changes, the absence of hillslope diffusion in M1 ($k_{hl} = 0$) leads to the largest surface roughness (Fig. 3a). When hillslope diffusion is included, the landscapes in models M2, M3, and M4 are smoother than those in model M1 (Fig. 3b-d). For models M2 and M4, doubling both the diffusion and erosion coefficients

175   reduces both the mean elevation and the mean surface roughness by a factor of ~2. For models M2 and M3, doubling only the diffusion coefficient reduces the surface roughness by ~15% and increases the mean elevation by ~20%. For models M3 and M4, doubling the erosion coefficient alone reduces the mean elevation by a factor of more than 2.

Stronger diffusion smooths local slopes, but it also causes the net deposition of sediment into valleys. To maintain equilibrium with a constant uplift rate, the river needs to erode not only the uplifted bedrock but also the additional materials from the

180   hillslope. This process forces the channels to become steeper to gain the necessary power to cut through the combined load of bedrock and sediment (Litwin et al., 2025). As stronger diffusion widens valley spacing and forces channels to steepen, the total relief and mean elevation of landscapes increase.

[Figure]

**Figure 3. Hillshade maps showing erosion and deposition rates resulting from hillslope diffusion at the end of Stage 1 and the end of Stage 2 for models M1 (a), M2 (b), M3 (c), and M4 (d). Each model differs in hillslope diffusion coefficients ($k_{hl}$) and erodibility values ($k_d$). Blue areas indicate deposition, while red areas represent erosion. Color bar values indicate depositional (positive) and erosional (negative) rates (mm/yr). Numbers below each map display the mean elevation (black), drainage density (blue), and roughness (red). Dashed lines on maps at the end of Stage 1 denote the divides. The divides in Stage 2 are similar to those in Stage 1 and are not marked in this stage.**

**3.2 Impact on river channel response**

185    To explore channel responses to changes in rainfall or uplift rates under various ratios of hillslope diffusion to erodibility, we analyze the trunk stream of the western basin, including the evolution of erosion and deposition, as well as the evolution of the longitudinal channel profile. Although we present results only from the western basin, we have verified that both drainage basins exhibit similar evolutions.

**3.2.1 Null-case control (Model M1, $k_{hl}$ = 0)**

190    To isolate the impact of hillslope diffusion, we first present the results from model M1, which has a diffusion coefficient of zero and serves as our null case (Fig. 4). This model illustrates the baseline landscape response when driven purely by riverine processes, showing the development of a standard migrating knickpoint. By establishing this null case, we can then clearly distinguish the critical role of hillslope diffusion in landscape evolution in models M2, M3, and M4.

In the absence of hillslope diffusion, when the rainfall rate decreases or the uplift rate increases, the trunk stream rises gradually,

195    and the slope increases from the river mouth. A positive knickpoint and an erosion wave develop at the river mouth and migrate upstream (Fig. 4a and d). The downstream channel reaches a steady state first, with no further changes in elevation or slope.

Conversely, when the rainfall rate increases or the uplift rate decreases, the channel's elevation and slope decrease. A negative knickpoint and an erosion wave develop at the river mouth and migrate upstream (Fig. 4b and c). Once the erosion wave reaches the headwaters, the knickpoint disappears, and the entire channel returns to a new steady state. Notably, within 1-2

200 Myrs of the change in rainfall or uplift rates, the channel elevation at the headwaters changes, but the slope remains nearly constant (Fig. 5 a1-3 and Fig. 6 a1-3). As the erosion wave approaches the headwaters, the channel slope increases or decreases monotonically and eventually stabilizes.

[Figure]

205 **Figure 4. Longitudinal profiles of the trunk stream after changes in rainfall or uplift rates in model M1 (no hillslope diffusion). The changes occur at 25 Ma, affecting the steady state trunk stream in blue.**

[Figure]

When the hillslope diffusion is absent (model M1), the slope of the trunk stream headwaters remains nearly constant for 1-2 Myrs following a change in either uplift or rainfall rate (Fig. 5 a1-3 and Fig. 6 a1-3). The slope then increases monotonously after the arrival of the upstream migrating erosion wave (Whipple and Tucker, 1999).

Figure 5. Evolution of trunk stream slope following an increase in uplift rate. (a1-d1) Longitudinal slope profiles of the trunk stream at selected time steps (colored lines), with each subplot corresponding to a model (M1-M4). Black rectangles indicate the headwater regions. (a2-d2) Enlarged views of the headwater areas, corresponding to the boxed regions in (a1-d1). (a3-d3) Temporal evolution of the mean channel slope in the upper ~800 m of the trunk stream, capturing the dynamic slope response across model runs. Dashed vertical lines mark the timing of the uplift rate increase (25 Ma).

[Figure]

**Figure 6. Evolution of trunk stream slope following a decrease in rainfall rate. (a1-d1)** Longitudinal slope profiles of the trunk stream at selected time steps (colored lines), with each subplot corresponding to a model (M1-M4). Black rectangles indicate the headwater regions. **(a2-d2)** Enlarged views of the headwater areas, corresponding to the boxed regions in (a1-d1). Grey bands indicate the regions where the transient slope change reversal occurs. **(a3-d3)** Temporal evolution of the mean channel slope in the upper ~800 m of the trunk stream, capturing the dynamic slope response across model runs. Dashed vertical lines mark the timing of the rainfall rate decrease (25 Ma).

**3.2.2 Diffusion-enabled models (M2–M4)**

In contrast, when hillslope diffusion is present (models M2, M3, and M4), we observe major differences in the evolution of headwater channel slope following changes in uplift and rainfall rates. An increase in uplift rate leads to a monotonic slope increase in the headwaters (Fig. 5 b1-3, c1-3, and d1-3). In contrast, a decrease in rainfall rate triggers a "*transient slope change reversal*", a phenomenon we define as a non-monotonic adjustment where the headwater channel slope initially changes in the opposite direction of its final steady state. This is observed as a transient slope decrease followed by a subsequent, long-term increase (Fig. 6 b1-3, c1-3, and d1-3). The opposite pattern occurs when the rainfall rate increases: a temporary slope increase is followed by a decrease. We do not find a distinct threshold for the initiation of the transient slope change reversal; rather, it is present whenever hillslope diffusion is active (Pe < ∞). The primary control on the reversal is its magnitude and persistence, which vary continuously with Pe. Our results show that landscapes with lower Pe values, where

hillslope diffusion is more dominant relative to channel incision, exhibit more pronounced and persistent reversals. For example, model M3, which has the lowest Pe, shows a reversal that persists longer and extends over a longer channel segment compared to other models (Fig. 6 c1-3). Although models M2 and M4 share the same Pe value, the larger $k_{hl}$ and $k_d$ values of model M4 halve both the diffusion and advection timescales relative to model M2. Consequently, the transient slope change reversal persists longer in model M2 in time, even though the non-dimensional dynamics are identical (Fig. 6 b3 and d3).

**4 Discussion**

**4.1 Mechanism of transient slope change reversal**

To better understand the cause of the transient slope change reversal, we calculate the erosion rate for each grid cell 1 Myr after the disturbance and extract the erosion rate along the trunk stream for all models (Fig. 7). The transient slope change reversal is driven by differential erosion rates between the divide and adjacent areas.

In model M1, the erosion rates of the divide and its adjacent areas remain homogeneous following changes in rainfall and uplift rates (Fig. 7 a3). Similarly, in models M2, M3, and M4, an increase or decrease in uplift rate results in consistent erosion rates between the divide and adjacent areas (red and orange profiles in Fig. 7 b3, c3, and d3). The surface uplift rate is defined as the difference between the uplift and erosion rates. Given the spatial uniformity of uplift rates, equal erosion rates at the divide and its adjacent areas result in identical surface uplift rates, preventing transient slope change reversals (black and red profiles in Fig. 8).

In contrast, following a decrease in rainfall rate in models M2, M3, and M4, the erosion rate of the divide exceeds that of adjacent downstream areas (green profiles in Fig. 7 b3, c3, and d3). This difference in erosion rate directly causes the surface uplift rate of the divide to be lower than that of adjacent downstream areas, resulting in a temporary decrease in the channel slope at the divide and, therefore, triggering a transient slope change reversal (green profile in Fig. 8). Conversely, following an increase in rainfall rate, the erosion rate of the divide is lower than in adjacent areas (blue profiles in Fig. 7 b3, c3, and d3), causing a temporary slope increase at the divide and again triggering a transient slope change reversal (blue profile in Fig.8). These findings suggest that rainfall changes distinctly influence divide erosion patterns, with spatial contrasts in erosion rate playing a key role in driving transient slope responses.

[Figure]

**Figure 7. Erosion rates (mm/yr) per grid cell, calculated over 1 Myr following (a1-d1) a decrease in rainfall rate and (a2-d2) an increase in uplift rate. Blue lines in (a1-d1) and (a2-d2) represent trunk streams, and dashed lines mark divides. (a3-d3) Longitudinal erosion profiles along trunk streams, with grey bands indicating the regions where the transient slope change reversal occurs.**

[Figure]

**Figure 8. Schematic diagram of the longitudinal profile of the channel in a steady state (black line) or a transient state after changes in rainfall or uplift rate. The grey band indicates the region where the transient slope change reversal occurs.**

The transient slope change reversal is driven by a disequilibrium between the hillslope diffusion timescale and the channel advection (incision) timescale. Pe quantifies the ratio of these two timescales. Following a change in rainfall rate, the advection timescale, which is inversely related to incision efficiency, adjusts almost instantaneously. In contrast, the diffusion timescale, governed by topography, does not (Clubb et al., 2019). This abrupt shift in their ratio (i.e., the change in Pe) creates a lag and drives the transient behavior at the headwaters. For instance, following a decrease in rainfall rate, the advection timescale lengthens (river incision becomes less efficient) due to lower discharge (Mitchell, 2020; Montgomery et al., 2000). However, sediment continues to diffuse from divides to channels at a rate set by the pre-existing topography (i.e., the diffusion timescale is initially unchanged). This imbalance causes the rate of sediment supply from hillslopes at the headwaters to exceed the rate of sediment removal by rivers, reducing the channel slope temporarily and causing a transient slope change reversal. As the channel adjusts and the erosion wave migrates upstream, this reversal gradually disappears.

In contrast, a change in uplift rate uniformly raises the entire landscape without immediately affecting the efficiency of diffusion and incision. Because both the divide and its adjacent areas experience similar erosion conditions under constant discharge, no transient slope reversal occurs.

Notably, a lower Pe value amplifies the imbalance between sediment supply from hillslopes and removal by rivers. This enlarges the zone where divide erosion rates differ from downstream areas. Therefore, the transient slope change reversal persists over a longer channel segment and for a longer duration, as observed in model M3 (Fig. 6 c2 and c3). In contrast, increasing Pe enhances river incision, which reduces the relative influence of diffusion. This leads to a shorter channel segment experiencing transient slope change reversal and a shorter duration of the transient response in model M4 (Fig. 6 d2 and d3).

In summary, the transient slope change reversal results from the competition between incision and diffusion following a change in rainfall. This reversal disappears as the erosion wave gradually approaches the divide area, and the landscape returns to a steady state where the erosion rate is spatially uniform.

**4.2 Field and analytical approaches for detecting transient reversals**

Transient slope change reversals could be identified using slope-area analysis or χ analysis. Both methods rely on the stream power model, which describes the relationship between channel slope and drainage area as a power function (Flint, 1974). For a river channel in a steady state, plotting log slope against log area yields a straight line. However, in cases of transient slope change reversals, this relationship may deviate from linearity. While slope-area analysis can be sensitive to data noise (e.g., DEM inaccuracies), χ analysis reduces this influence through an integral approach (Royden and Taylor Perron, 2013; Perron and Royden, 2013). For steady-state rivers, χ should also correlate linearly with elevation, whereas nonlinear χ-elevation relationships may indicate transient slope change reversals. In our models, a decrease in rainfall rate produces a localized flattening at high χ (headwaters), directly reflecting the transient slope-change reversal (Fig. 9). By contrast, in uplift-driven transients the χ-elevation profile bows downward at low χ, while the high-χ (headwater) segment remains straight and is simply translated upward. However, χ-elevation analysis has limitations: it requires a steady-state baseline profile to distinguish

different types of disturbances. Therefore, χ-elevation is best used in concert with additional information, such as independent
300   erosion-rate measurements, to robustly identify and attribute transient slope-change reversals.

[Figure]

**Figure 9. χ-elevation profiles of trunk streams in model M3 under three conditions: following an uplift rate increase (green), following a rainfall rate decrease (orange), and steady-state (light blue). The three grey dashed lines are parallel reference trends. χ-elevation profiles are calculated using a reference concavity index ($\theta_{ref}$) of 0.4.**

Transient slope change reversals could also be identified by investigating the erosion rate. One approach to quantify erosion rates is using cosmogenic nuclides, particularly radionuclides like $^{10}Be$ and $^{26}Al$ (e.g., Balco et al., 2008; Gosse and Phillips,
305   2001; Lal, 1991; Muzikar, 2009). These nuclides are produced in surface minerals by cosmic ray interactions, with production rates decreasing exponentially with depth due to cosmic ray attenuation (Dunai, 2010; Lal, 1991). Cosmogenic nuclide concentrations increase as a surface remains exposed to cosmic rays (Ivy-Ochs and Kober, 2008). In contrast, in rapidly eroding areas, nuclide concentrations remain low due to the continuous removal of surface materials.

By mapping nuclide concentrations, spatial patterns in erosion rates could be linked to rainfall or uplift changes. For instance,
310   if the erosion rate is relatively uniform around the divide area, it may suggest a transient response driven by tectonic events. Conversely, if nuclide data indicate that erosion rates are larger at the divide relative to downstream areas, then recent drainage reorganization may be related to a decrease in rainfall rate. Thus, cosmogenic nuclide measurements provide a valuable tool to distinguish between climatic and tectonic drivers of landscape change.

**4.3 Model limitations**

315   In this study, we aim to explore the first-order impact of hillslope diffusion and river incision on landscape and consider a landscape evolving under the action of hillslope diffusion and river incision only. While the linear diffusion model is a common starting point, we acknowledge that it does not capture nonlinear processes, such as those driven by shallow landslides, which can become significant on steeper slopes (e.g., Jiménez-Hornero et al., 2005; Martin, 2000; Roering et al., 1999). Furthermore,

our model does not account for potential feedback between climate and the diffusion coefficient itself. In natural settings, the hillslope diffusion coefficient can vary with climatic conditions via processes such as frost-crack weathering, and near-surface processes such as soil saturation, and root growth (Braun, 2018; Perron, 2017; Bogaard and Greco, 2015; Andersen et al., 2015; Gabet and Mudd, 2010; Gabet, 2000). Considering this feedback could introduce additional complexity. For instance, an increase in rainfall rate could increase the hillslope diffusion coefficient through higher soil moisture (Perron, 2017), potentially amplifying the transient slope change reversal. Conversely, a decrease in rainfall rate could decrease the hillslope diffusion coefficient and dampen the reversal. Future work could explore the parameter space where these feedbacks become significant.

In addition, our use of a detachment-limited stream power model simplifies the complexities of sediment flux. The "transient slope change reversal" we observe is fundamentally a result of a disequilibrium between hillslope sediment supply and the channel's transport capacity following a change in rainfall. A more complex model incorporating sediment transport dynamics (a "transport-limited" or "mixed" model) would likely modulate the magnitude and duration of this reversal.

**5 Conclusion**

Changes in rainfall and uplift rates induce different responses in the channel slope at the headwaters, with hillslope diffusion playing a crucial role in mediating these processes. When the rainfall rate changes, hillslope diffusion interacts with river incision to generate transient spatial variations in erosion around the divide area, leading to transient slope change reversals at the headwaters. In contrast, changes in uplift rates result in spatially uniform erosion across the divide area, preventing such reversals. Identifying these reversals from river profiles or erosion rate estimates at different locations could help determine the driving force behind landscape adjustments. A high hillslope diffusion coefficient increases both the duration and spatial extent of these reversals along the river profile. In contrast, higher erodibility enhances river incision and diminishes the role of diffusion, reducing these reversal effects.

Our findings provide new insights into how rainfall and tectonic forcing reshape landscapes over time. By investigating the interaction between diffusion and incision, we show that the transient variations in channel profiles, particularly near the divide, provide potential markers for interpreting past landscape evolution and deciphering the complex interplay between tectonic uplift and climatic variability.

**Code and data availability.** Version 2.2.0 of Badlands used for the landscape and sedimentary evolution modeling is preserved at https://doi.org/10.5281/zenodo.1069573 (Salles & Howson, 2017), available via GNU General Public License v3.0 and developed openly at https://github.com/badlands-model/badlands.

**Author contributions.** YZ designed and ran the simulations, analyzed the results, and wrote the manuscript. PR contributed to the result analysis and manuscript revision. TS developed the model code and contributed to the manuscript revision.

**Competing interests.** The authors declare that they have no conflict of interest.

**Acknowledgments.** The first author gratefully acknowledges the financial support from the China Scholarship Council (CSC)
and the School of Geosciences at the University of Sydney. We thank the three anonymous reviewers for their constructive
comments and suggestions, which helped to improve this manuscript.